# Infrared spectroscopic laser scanning confocal microscopy for whole-slide chemical imaging

Kevin Yeh [1,2], Ishaan Sharma [1], Kianoush Falahkheirkhah [1,3], Matthew P. Confer [1], Andres C. Orr[1], Yen-Ting Liu [1,4], Yamuna Phal [1,4], Ruo-Jing Ho[1,2], Manu Mehta [2], Ankita Bhargava[5], Wenyan Mei[6,7,8], Georgina Cheng [8,9], John C. Cheville[10] & Rohit Bhargava [1,2,3,4,8,11,12] ✉

Chemical imaging, especially mid-infrared spectroscopic microscopy, enables label-free biomedical analyses while achieving expansive molecular sensitivity. However, its slow speed and poor image quality impede widespread adoption. We present a microscope that provides high-throughput recording, low noise, and high spatial resolution where the bottom-up design of its optical train facilitates dual-axis galvo laser scanning of a diffraction-limited focal point over large areas using custom, compound, infinity-corrected refractive objectives. We demonstrate whole-slide, speckle-free imaging in ~3 min per discrete wavelength at 10× magnification (2 μm/pixel) and high-resolution capability with its 20× counterpart (1 μm/pixel), both offering spatial quality at theoretical limits while maintaining high signal-to-noise ratios (>100:1). The data quality enables applications of modern machine learning and capabilities not previously feasible – 3D reconstructions using serial sections, comprehensive assessments of whole model organisms, and histological assessments of disease in time comparable to clinical workflows. Distinct from conventional approaches that focus on morphological investigations or immunostaining techniques, this development makes label-free imaging of minimally processed tissue practical.

Label-free chemical imaging (CI) techniques are a powerful complement to light microscopy in that molecular and morphological contrast is derived intrinsically from the sample rather than through attaching purpose-designed fluorescent probes to biomolecules of interest or using non-specific dyes. Mid-infrared (IR) spectroscopic imaging promises the strongest optical molecular sensitivity[1], in particular, arising from light absorption at mid-IR frequencies being resonant with molecular vibrational modes. Sophisticated analyses of

[1]Beckman Institute for Advanced Science and Technology, University of Illinois at Urbana-Champaign, Urbana, IL 61801, USA. [2]Department of Bioengineering, University of Illinois at Urbana-Champaign, Urbana, IL 61801, USA. [3]Department of Chemical and Biomolecular Engineering, University of Illinois at Urbana-Champaign, Urbana, IL 61801, USA. [4]Department of Electrical and Computer Engineering, University of Illinois at Urbana-Champaign, Urbana, IL 61801, USA. [5]University of Illinois Laboratory High School, Urbana, IL 61801, USA. [6]Department of Comparative Biosciences, College of Veterinary Medicine, University of Illinois at Urbana-Champaign, Urbana, IL 61802, USA. [7]Carl R. Woese Institute for Genomic Biology, University of Illinois at Urbana-Champaign, Urbana, IL 61801, USA. [8]Cancer Center at Illinois, University of Illinois at Urbana-Champaign, Urbana, IL 61801, USA. [9]Carle Health, Urbana, IL 61801, USA. [10]Department of Laboratory Medicine and Pathology, College of Medicine and Science, Mayo Clinic, Rochester, MN 55905, USA. [11]Department of Chemistry, University of Illinois at Urbana-Champaign, Urbana, IL 61801, USA. [12]Department of Mechanical Science and Engineering, University of Illinois at Urbana-Champaign, Urbana, IL 61801, USA. ✉e-mail: rxb@illinois.edu

these data, especially in conjunction with recent ground-breaking progress in artificial intelligence (AI), have dramatically expanded scientific and biomedical opportunities with chemical imaging[2]. This combination can open the avenue for rapid and comprehensive spatial and molecular profiling without the need for prior knowledge or human supervision. However, this exciting potential remains locked by a rate of IR chemical imaging data acquisition that is too slow and quality of data that is too poor to be of routine use. The first limitation arises from IR optical systems needing to be different from those used for light microscopy since typical glass blocks effectively all IR light. Second, the full mid-IR spectral bandwidth (~12 μm) is more than 30-fold larger than visible (~0.32 μm) necessitating microscopy with all-reflective optics. These microscope designs have remained essentially unchanged since the early IR microscopes developed prior to 1950[3,4]. Third, analyses of biomedical samples[5] have greatly relied on making trade-offs between spatial localization and spectral data quality[6]. Despite a massive expansion in the capabilities[7,8] and applications[9,10] of IR microscopy, the speed of data acquisition does not readily provide the large and diverse data sets that make techniques like deep learning effective. A re-imagining of the IR microscope was necessary to address these limitations by designing custom optical components, optimizing the optical train with modern design and components, and integrating fast acquisition and analysis.

Pre-built or commercially available optical assemblies—including high magnification compound refractive objectives, tube lenses, and scan lenses—that are well-corrected for a broad mid-IR spectral range are currently unavailable. Hence, the challenge was to design, fabricate, evaluate, and integrate each of these components. While one of the advantages of an all-reflective design was compatibility with the broadband, but weak, thermal source, our design is timely with the emergence of broadly tunable but narrowband-emitting quantum cascade laser (QCL) technology[11] whose efficacy has been shown for IR microscopy[12–16]. In these microscopes, the high intensity of the QCL source provides an advantage in signal and greater spatial localization as well as allows an increase in imaging speed by discrete frequency (DF) measurements[17]. However, the limitations of microscope designs and/or spatial coherence preclude rapid acquisition of high signal-to-noise ratio (SNR) that is critical for biomedical analysis[18]. Artifact-free data, in contrast, are obtainable by conventional Fourier transform IR (FT-IR) imaging systems. FT-IR imaging systems necessarily record a wide bandwidth to gain their large multiplexing and SNR advantage but consequently also result in low acquisition speed. Overcoming these limitations, our goal here was to design a microscope that exceeds the imaging speed and quality of current state-of-the-art QCL microscopes in DF conditions as well as surpasses the efficiency of FT-IR microscopes under hyperspectral conditions.

Here we report on the design, its physical realization, and demonstrations in achieving the performance metrics for serving as a rapid biomedical analysis method. Our strategy centered on designing a confocal laser scanning microscope (LSM) that is well-corrected for third-order optical aberrations across the large mid-IR spectral range.

## Results

### Design and performance evaluation of IR-LSM

LSMs are often reported for rapid, high-resolution imaging, where small oscillating mirrors quickly sweep a beam through a complex series of optics designed to precisely maintain a tight focal volume across the sample. Our IR-LSM design follows the same strategy, consisting of a series of fully custom compound optics designed in tandem—namely, a 4× magnification scan lens (SL), a tube lens (TL), and two interchangeable infinity-corrected objectives (OBJ) for 10× and 20× magnification with numerical apertures (NA) of 0.4 and 0.8 respectively—all corrected for mid-IR wavelengths that are 10-fold longer and of a 10-fold wider bandwidth than typical design specifications for the visible spectrum. The designed system is detailed

schematically in Fig. 1a with a glossary of abbreviations presented in the Supplementary Information. It uses an assembly of four external cavity QCLs, emitting a narrow-band beam tunable across the molecular fingerprint spectral range. A detection pinhole passes only the transflected light conjugate to a near-diffraction limited focal volume on the sample plane while rejecting stray reflections from optical and mounting surfaces that manifest as ghost images or lens flare. The individual optical assemblies are designed by forgoing the use of common, but IR-opaque, silica glasses. Instead, we guided an optimizer to continuously model and search through thousands of combinations of materials and surface parameters by fine tuning individual constraints bounding the search space and weighing key performance metrics (Supplementary Fig. S1 and Supplementary Table S1). Additional considerations included form factor restrictions relating to end-user practicality and fabrication feasibility, analyzing tolerance sensitivity to maximize the probability of achieving designed results, and ensuring that costs did not become unduly high. From a balance of these factors, we selected ~10 leading designs and further iterated each depending on current manufacturing capabilities. After optimization, the final selections were custom fabricated (Fig. 1b, Supplementary Fig. S2 and Supplementary Table S2) in multiple iterations to ensure performance as per design. The two objective assemblies we obtained offer a balance between common coverage and resolution, requiring materials indicated by the color-coded scheme. While these assemblies have considerably more elements than most off-the-shelf IR optics, as necessary to achieve high focusing power over a wide spectral and spatial range, the key to this design is that it is still relatively simple compared to most visible microscope assemblies, and thus practical to realize. The optimized surface parameters and theoretical performance curves of these designs are presented in the Supplementary Information (Supplementary Figs. S3–6).

After fabrication, we incorporated these optical assemblies into a custom-built IR microscope frame following the schematic in Fig. 1a and illustrated in Supplementary Fig. S7. Notably, the design of the objectives was also an opportunity to match and optimize the light path through the LSM to maximize throughput, eliminate stray light and match components for close to ideal performance. We first evaluate the spatial and spectral performance of this microscope in Fig. 1c(i)–(ii) with calculations of these benchmarks described by Supplementary Table S3. A 100% line is the standard measure of spectral fidelity, showing a uniform response across the bandwidth. The spatial noise in Fig. 1c(ii) reflects the SNR of the measurement, showing a detector cutoff effect (lower wavenumber side) as well as higher noise for the IR-LSM that is associated with low-power transitions of the multi-module laser system. While FT-IR systems have uniform noise across the bandwidth due to interferometric recording, we compare the spectral and spatial noise respectively, recorded with the typical number of coadditions for common experimental conditions, and find that the IR-LSM is capable of equal or lower noise than FT-IR microscopy with lower levels of signal averaging (coadditions) required. This performance directly impacts experimental time, as shown in Fig. 1c(iii)–(iv), demonstrating that IR-LSM has higher imaging throughput (pixels/time) than FT-IR microscopes. Note that the IR-LSM experimental time scales linearly with the number of bands required. While we benchmark with 100 DF bands for comparative purposes here, most biomedical studies use ~10 bands and offer faster speeds in practice. The SNR of this data, moreover, is better than state-of-the-art FT-IR microscopes. Notably, the performance of IR-LSM is not normalized for SNR or by sampling volume. As evident in Fig. 1c(iii)–(iv), 16-coaddition FT-IR images are comparable in SNR to IR-LSM 2-coaddition images despite taking over 4-fold longer per megapixel (MPx) per 100 bands.

Though IR imaging systems have been theorized to provide high spatial localization of spectral signatures[8,19,20], this performance has been difficult to achieve without significant increase in experimental

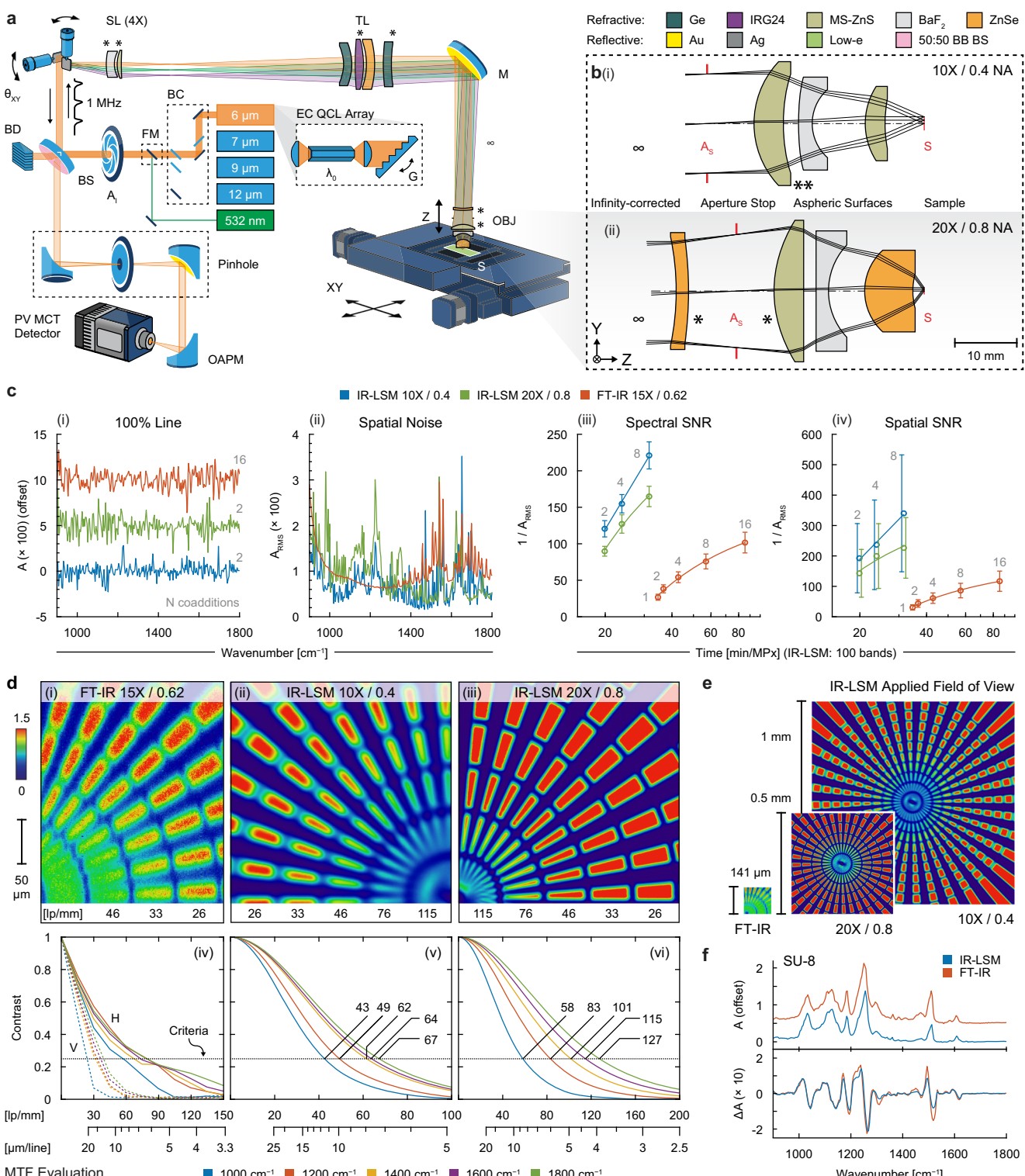

**Fig. 1 | Design overview and technical evaluations of label-free IR-LSM.**
**a** Schematic is based on a galvanometer scanning confocal design with refractive optical assemblies custom-built for broadband performance and coupled to an EC-QCL array spanning the mid-IR spectral range. **b** Cross-sections of the interchangeable 10× and 20× objectives with design specifications are presented in the Supplementary Information (Supplementary Figs. S3 and 4) in conjunction with the tube and scan lens (Supplementary Figs. S5 and 6). Technical comparisons between FT-IR and IR-LSM configurations including: (**c**) system noise and SNR calculations (detail in Supplementary Table S3) with evaluations per (i) single pixel 100% lines in absorbance, (ii) spatial noise per wavenumber, (iii–iv) spectral and spatial noise obtained over time (IR-LSM: 100 bands, FT-IR: 4 cm⁻¹ spectral resolution) presented as mean values ± s.d. (*n* = 1000); **d** spatial resolution demonstrated with (i) subjective images of a Siemens star target, and with (iv–vi) objective MTF curves evaluated respectively at discrete wavenumbers using a slant-edge target; **e** applied field of view; and **f** single pixel measurements of absorbance and derivative spectra of SU-8 photoresist deposition. A comprehensive glossary of terms is in the Supplementary Information.

times and lower SNRs with the prevalent strategy of modifying existing microscopes. The ab initio design of IR-LSM provides the sought-after resolving power and image quality as well as high SNR. Siemens star targets were imaged to quantify performance, with a single quadrant shown in Fig. 1d(i)–(iii) for comparisons. Notably, even our 10×/0.4 NA configuration outperforms the standard FT-IR imaging configuration in image quality, emphasizing that it is not just the nominal NA of the objective lens but the entire optical path that needs to be optimized. The high-NA IR-LSM configuration also provides a higher image quality without compromising SNR. The concentric rings of features patterned with known circumferential frequencies quantifies resolving power by determining the spatial frequency where the intensity modulation in the image falls below a predetermined threshold, commonly ~25% to approximate the Rayleigh criterion. This ability of an optical system to measure contrast is described as the modulation transfer function (MTF), which we calculate using the slant-edge method (ISO 12233) as shown in Fig. 1d(iv)–(vi). First, note that the resolution of the FT-IR imaging system in transflection sampling configuration is not symmetrical as part of the back aperture of the objective is used for illumination via a folding mirror. Hence, this asymmetric performance is quantified with separate MTF curves for the horizontal and vertical directions. In addition to poor vertical resolution, a loss of mid-frequency response can be observed in the MTF curves due to the central obscuration of the reflective objective. We do not see any such degradation in IR-LSM, which is also operated in transflection geometry, due to its use of refractive optics where light is illuminated and collected using the full aperture. Our design also minimized other difficulties typically associated with refractive IR designs. Despite the high number of refractive elements, the system exhibits negligible internal reflections (ghosting and lens flaring) even though the risks of such effects are ordinarily increased when using a coherent laser source. Lastly, the LSM design is also not susceptible to speckle.

The performance of these two designed objectives is shown (Supplementary Fig. S1c, d) and key specifications are compared (Supplementary Table S1) against FT-IR imaging, and other QCL-based[14,21,22] high performance microscopy systems. Most apparent from a method perspective, high-fidelity performance is maintained for ~12-fold and ~51-fold larger FOV (0.8 and 0.4 NA objectives respectively) compared to FT-IR imaging, as shown in Fig. 1e. Although the use of widefield measurements can provide larger fields of view, this design is neither limited by the size of the detector array nor by speckle or shot noise when an incident beam illuminates a large sample area[10,23]. Using the full intensity of the beam on each pixel allows high-fidelity measurements (Fig. 1f) that correlate well with FT-IR reference spectra and provide high SNR rapidly, making imaging fast. The LSM design is also ~10-fold faster than stage scanning[22] which was already more capable than the most efficient widefield imaging[21]. While interferometric techniques require mathematical transforms of large volumes of data to preclude rapid imaging, DF imaging using array detectors is fast but widefield laser illumination inevitably results in coherent interference and laser speckle[14] precluding analytical methods that require high precision or involve sensitive AI algorithms. Without these limitations, IR-LSM allows the realization of the unique potential of IR imaging in rapidly and comprehensively imaging biomedical samples.

## Characterizing complete animal models for organism-scale studies

IR-LSM can increase the area of investigation, provide high spatial detail, and increase speed for imaging single organisms or organs, currently limited primarily by beam steering speeds, enabling statistically relevant sample sizes thereby making the discovery potential of IR imaging accessible. Zebrafish are commonly used in medical research, for example allowing cost-effective testing of toxicity and

efficacy of potential therapeutics[24] that are vital for the current fast-fail approach in drug design[25]. Despite its chemical specificity, the long acquisition time IR microscopy has precluded its use in high-throughput model organism screening and limits the ability to examine whole organisms or organs. We illustrate these potentials of IR imaging by analyzing sections of zebrafish embryos, which are an ideal test vector because of their genetic similarity with humans, relative low-cost husbandry, small size, and abundant availability[26]. Embryo development and growth of the whole organism can be recorded (Fig. 2a), revealing details at a level comparable to brightfield images of H&E stains (Fig. 2b), but with additional mid-IR spectral information implicating chemical differences associated with various biological systems. Label-free IR imaging also allows for wild-type animals to be used instead of transgenic fluorescent zebrafish, allowing more economical research and faster and more cost-effective pivots when needed. In contrast to the embryonic stages (~2 mm), adult zebrafish (~25 mm), present a more realistic model as they have fully formed organs. In Fig. 2c, d, IR chemical maps for a whole section of wildtype zebrafish reveal all internal organs and their chemical differences, both coverage and detail which were not previously possible. We do not analyze the full chemical changes and compositional differences in this study but do show some of the key spectral data in Fig. 2e from regions indicated. The database of spectral features for each part of the fish allows not just a baseline for comparisons in disease or development but also for applications in which traditional techniques are insufficient. In Fig. 2f we show images of zebrafish with polymeric particles highlighted to mimic the presence of microplastics in marine species. Notably, plastics can be detected with this background of biological tissue easily. In comparison, FT-IR imaging techniques would not be able to survey samples of this size in reasonable time, and coherent artifacts in widefield DFIR instruments would not have the sensitivity for identification. LSM data allows both a recognition of the presence of polymer particles as well as spectral analyses to ascertain their identity.

## Stain-free histopathology for whole-slide chemical imaging, in two- and three-dimensions

The fast acquisition speeds enabled by IR-LSM allow IR whole-slide imaging (WSI), which is the current standard for most clinical diagnostics and research. Figure 3a shows high-resolution images obtained with the designed system and facile histologic recognition from the distinctive features of colon tissue in this rapid, non-destructive, and dye-free method. Although widefield cameras have been used to visualize large tissue sections rapidly, the images are affected by the tissue structure itself due to speckle arising from the coherent scattering of laser light[15]. One remedy commonly found in many scenarios involving laser projection, is to temporally randomize the laser speckle[27] and integrate for longer than the coherence length or decrease magnification such that the pixel size greatly exceeds the speckle size. Neither option contributes toward fast and high-fidelity imaging. Furthermore, the resolution enhancement of confocal over widefield detection across modalities is well-known. The quality of the data acquired, thus, is a major concern even beyond the metrics such as number of pixels or rate at which they are acquired. Analyses of optimal pixel sampling and integration time, or the trade-offs between multiplexed arrays and point scanning, are possible[28], but an assessment of the images makes it clear (Fig. 3a(i–iii)) that IR-LSM provides artifact-free data that has not previously been demonstrated by IR widefield imaging and, consequently, suffers from no degradation in spatial or spectral quality.

These reported images, generated from intrinsic chemical contrast in tissue, can identify key morphologic structures without any further need for segmentation while also demonstrating detail comparable to traditional H&E-stained images in Fig. 3b. Over decades, medical centers have gathered large archives of formalin-fixed,

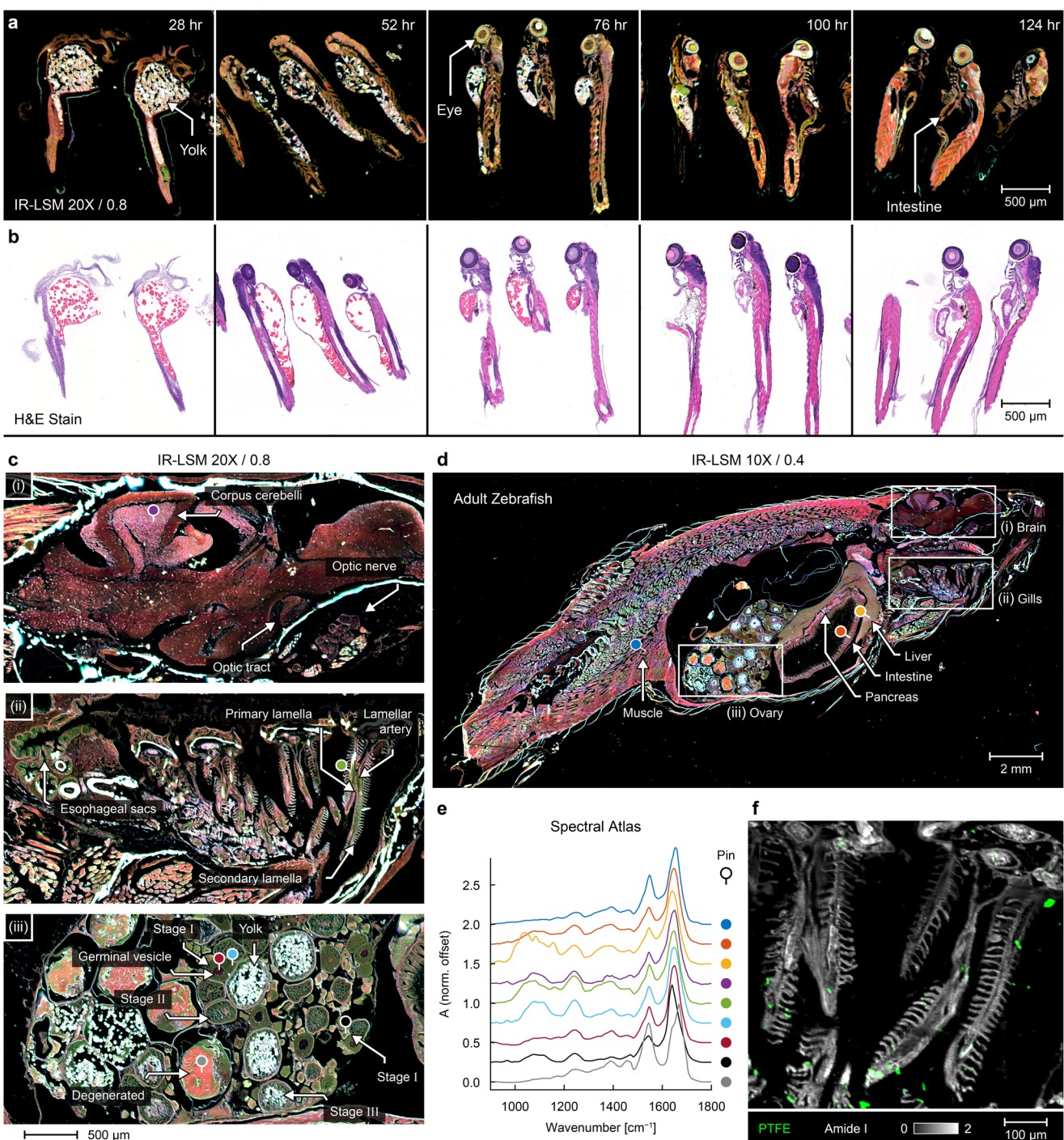

**Fig. 2 | Imaging sagittal sections of embryonic and adult zebrafish shown in false-color compositions of key IR absorption bands.** Embryonic zebrafish prepared at 28, 52, 76, 100, and 124 h post fertilization were imaged using (**a**) the presented IR-LSM instrument equipped with 20×/0.8 NA magnification, and (**b**) brightfield microscopy of H&E-stained sections of similarly prepared samples at the same stage. **c** Adult female zebrafish organ imaging was demonstrated with high-resolution insets of the (i) brain, (ii) gills, and (iii) ovary with the whole fish image presented in (**d**) and (**e**) fingerprint region spectra, at locations indicated by the pin symbol (from top to bottom), of tail muscle (navy), intestine (orange), liver (gold), cerebellum (purple), lamellar artery (green), primary growth stage I oocyte (teal) with germinal vesicle (burgundy), a small stage I oocyte (black), and degenerated oocyte (gray). **f** Contamination of PTFE microplastics in the zebrafish tissue sample with the Amide I absorbance band shown (grayscale) and detected PTFE signature at masked at 1213 cm⁻¹ and false-colored (green).

paraffin-embedded (FFPE) tissue samples that have provided potential for research discovery. Although stained images acquired using high-speed brightfield WSI scanners have digitized the morphology of these repositories, a wealth of additional chemical information was inaccessible and could not be evaluated due to the inadequate capacity of IR imaging technology. Though IR imaging has shown utility in adding prognostic information that adds capability to current morphological

imaging by measuring the microenvironment[29–31], the lack of IR-WSI has limited its assessment as a clinical diagnostic and prognostic tool. A typically accepted standard for WSI by optical microscopy is to scan a 15 mm × 15 mm area in minutes. Our designed system requires 2.8 min/band with the 10×/0.4 NA objective for 2 μm/px quality. Typically, less than 10 bands are needed for many applications in histopathology, thus bringing the time for scanning within range of time-

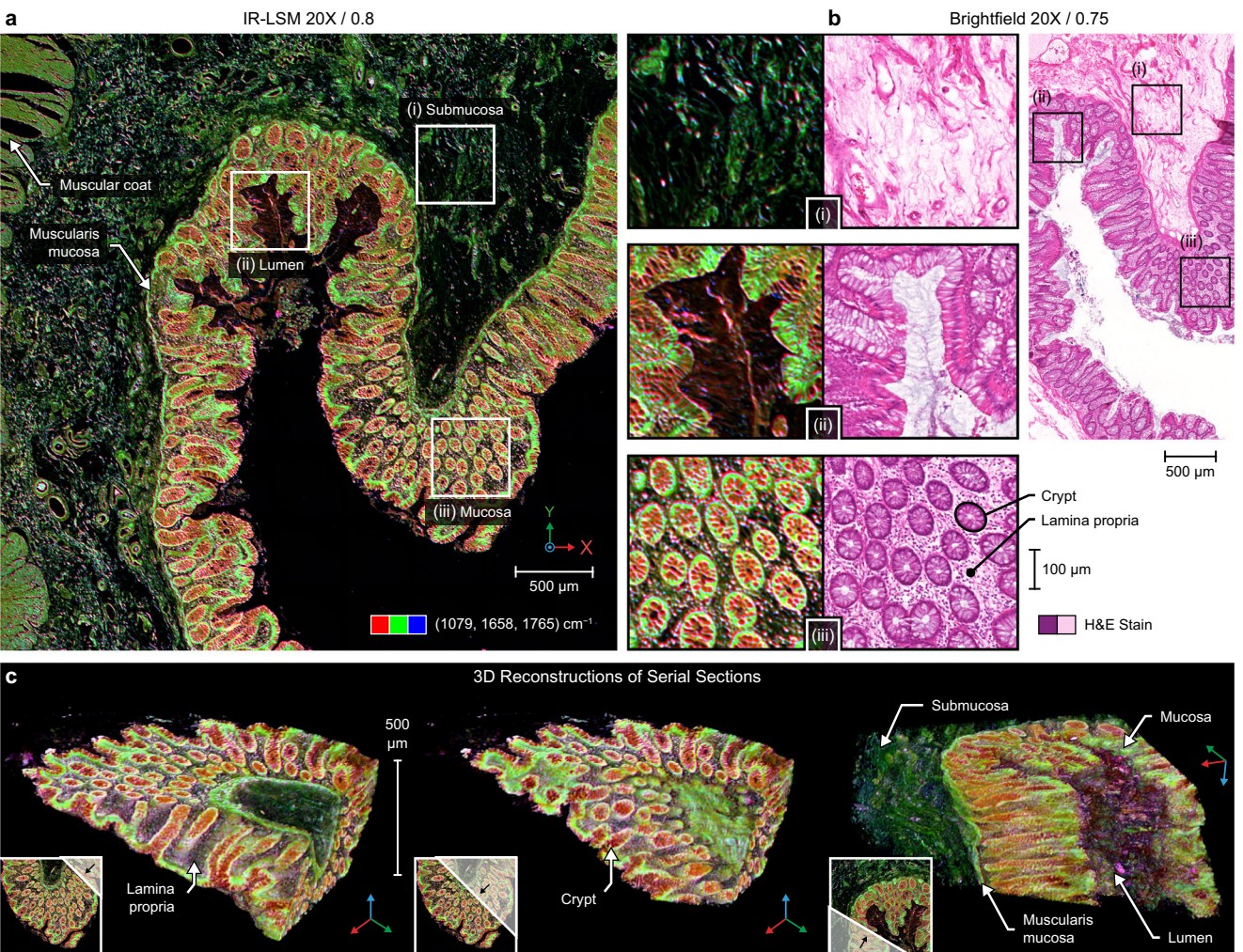

**Fig. 3 | Visualization of colon tissue under IR-LSM. a** False-color images reveal the distribution of phosphates and proteins in the tissue, where (i–iii) enlarged regions of submucosa, lumen and mucosa show comparable detail to (**b**) H&E-stained images of an adjacent tissue section observed under brightfield microscopy with corresponding regions as indicated. **c** Serial sections of the tissue block (100 slices, 5 μm thick) are imaged and computationally aligned to create a unique perspective rendering of the colon mucosa in 3D space. The corner insets show the cross-sectional position from which the corresponding view was generated from the original volumetric data, with the vantage angle as indicated by the black arrow.

sensitive histology tasks such as intraoperative tissue assessment, whose potential we demonstrate in the next section focused on potential for clinical diagnostics. For research purposes, the speed here allows for large-scale data recording that is necessary to assess extensive regions of the spectrum or acquire high volumes of data for sophisticated approaches such as deep learning. Notably, the tissue imaging capability here is 10-fold faster than the previous fastest laser scanning method[21,22], without compromising the SNR of the recorded data.

The data acquisition speed of IR-LSM not only enables extensive data recording but also interesting capabilities. Several studies have now shown the ability to record morphological data from three dimensions that can greatly improve understanding of disease processes[32,33]. The speed of IR-LSM allows the capability to create volumetric 3D reconstructions based on IR data, providing a path to study both the complex structural and correlated chemical changes during disease processes in three dimensions. Note that IR-LSM is still intrinsically a 2D imaging technique, requiring the sample to be serially sectioned, due to the very high extinction coefficient of tissue vibrational absorption in the mid-IR spectral region. Figure 3c allows the visualization of any cross section from this volumetric data, allowing an appreciation of the three-dimensional structure of crypts that may not be apparent in 2D sections.

Visible light sheet imaging offers unprecedented volumetric acquisition speeds[34] but needs staining; likewise, tissue clearing[35] methods allow exquisite structural imaging but removes lipids and other chemicals. 3D reconstructions with IR-LSM present a middle ground that is sensitive to a vast array of molecular signatures providing potential for discovery in the chemical domain and ability to measure subtle changes in the functional and microenvironmental parts of the tissue.

## Reducing time for clinical decision-making

While the previous application focused on archival quality tissue, the formalin fixation, embedding, and sectioning processes require significantly longer time (~day) than needed to scan resected tissue (~hour). Thus, we sought to examine whether IR-LSM can enable more rapid pathology. Samples can be prepared using a cryostat microtome to provide chemical-structural assessment from excision to image that is suitable for settings in which rapid assessment is needed, for example, intraoperative diagnoses. We demonstrate this capability by imaging and analyzing fresh frozen prostate tissues from surgical resections. These are fresh tissue samples that are flash frozen and sectioned on site, which better conserves chemical information[36], but requires imaging in <1 h when outside deep-freeze to prevent extensive degradation. Thus, the speed of the IR-LSM is

essential to examining fresh frozen tissue. The clinical standard is to prepare the tissue sample, stain with H&E and use optical microscopy for morphologic assessment by board-certified pathologists. In contrast, IR imaging uses machine learning techniques, using spatial-spectral data to make automated and objective assignments of disease. First, we imaged surgical resections and obtained annotations of the ground truth (normal and diseased epithelial cells) as shown in Fig. 4a, b for training the deep learning model. Our model is a U-Net (Fig. 4c) that segments the prostate samples into three important histological units: benign epithelium, malignant epithelium, and non-epithelial tissue (collectively, stroma). We illustrate the performance of our method under three different conditions (Fig. 4d–f), where raw IR images are evaluated by our model to ascertain the presence of benign epithelial cells in normal tissue, whereas malignant tumors are clearly identified in cancerous tissue, subjectively in agreement with the corresponding parallel H&E images (Fig. 4g, h). The performance of this model is calculated to have AUC ROC values of

0.932 (benign), 0.952 (malignant), and 0.968 (non-epithelium) estimated using 10,000 pixels for each class. Finally, to evaluate the performance of our approach for localizing different histological classes within the same tissue section, we assessed regions that contain both benign and cancerous cells. It is evident that rapid IR imaging coupled with machine learning can provide accurate segmentation of benign and cancerous tissue from frozen sections, without the need for stains or human assessment, thereby laying the foundation for streamlining and accelerating the current workflow to the point of surgical care. The benefits of the advances reported here are of more fundamental to the fledgling field of IR imaging for pathology as well. As deep learning approaches appreciate greater spatial details and textures[37], our instrument can improve data analysis and capabilities of IR imaging. Additionally, our use of unstained fresh frozen samples for imaging preserves the tissue structure and chemistry for further downstream analyses[38], which may include immunohistochemistry and/or sequencing.

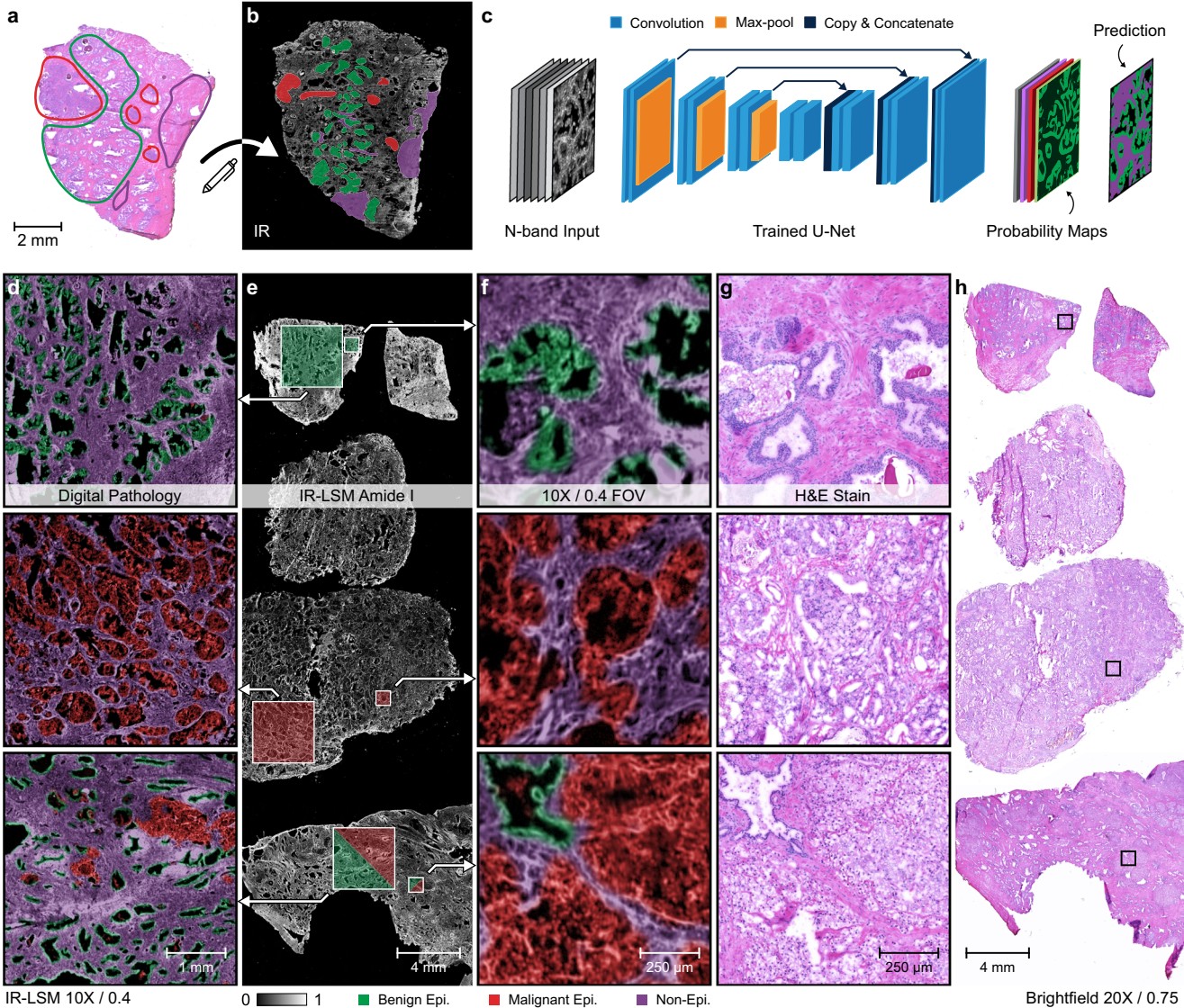

**Fig. 4 | Demonstration of prostate cancer detection in a preparation of fresh frozen tissue sections. a** Rough annotations guided by pathologists are (**b**) transferred from H&E images to IR images on consecutive sections. **c** A deep learning model is trained to predict epithelial (epi.) malignancy using multispectral IR images. **d** Validation is performed on unstained samples containing benign glands, malignant glands, and benign glands surrounded by malignant tumors, each imaged with IR-LSM and represented in each row. **e** Target ROIs are shown on the Amide I map, representing a full microscopy slide, with (**f**) indicative IR-LSM 10× magnified field of view simulating real-time screening. **g, h** Comparative images of H&E-stained adjacent sections of the same tissue sample are shown with corresponding ROIs marked.

## Discussion

Current options for spatial-spectral assessment in IR chemical imaging rely on widefield DF imaging designs that suffer from speckle, full spectral FT methods that offer high informational content but result in needlessly large data sets with low acquisition speed, or high-to-ultrahigh resolution DF photothermal methods that are also slow. The designed system, even with the 10× objective, matches the best resolving power of FT-IR imagers, offers a dramatically higher SNR for short scan times and is speckle-free. The performance with a 20× objective far exceeds the current state-of-the-art. Our design is complementary to several other emergent technologies in the form of hybrid microscopes that aim to offer different capabilities. Concurrent illumination of the sample with a visible probe beam, for example, takes advantage of optical microscopy, well-developed visible optics, sensitive detectors and modulation schemes to measure minute deflections and achieve resolution beyond the IR diffraction-limit[39–44]. Using atomic force microscopy (AFM) probes[45–50] offers even higher resolution down to the nanoscale whereas ultrasonic detection[51] offers the ability to image intact, thick samples. However, measuring (weaker) downstream effects of IR absorption on the sample with a second probe requires more complex instrumentation and a slow-down due to the need to modulate-demodulate. Only requiring a single laser, with nanoseconds long illumination to provide sufficient SNR for typical biomedical samples in our design offers a fundamentally faster route. As we show here, relatively simpler, well-designed instruments can achieve the WSI throughput needed in modern cell-tissue level analyses and data quality that is useful for many research and diagnostic uses. The key challenge that was addressed is that refractive systems are difficult to implement in IR, typically needing a trade-off between the correctable wavelength range, performance, and design simplicity which requires mitigation by more optical surfaces, unique materials, complex geometries, and specialized anti-reflection (AR) coatings. Together, these factors inevitably reduce throughput. The availability of modern QCLs ensures that a light throughput of ~35% through our large lens stack still offers high performance. Matching these factors in a point scanning design especially provides sufficient throughput, even for lasers that are of a moderate output as used here. The dominant advantage of reflective optical designs arises partly from their broadband achromatic compatibility, a performance standard that is effectively met by the design here.

Though the approach of direct detection using high-speed IR-sensitive photonic detectors was capable of large-area imaging and offered appropriate SNR[21] for all-digital pathology[22], a further speedup was needed to make WSI feasible. The 10× configuration provides a balance between image quality and speed that brings IR imaging into the realm of common microscopy techniques for WSI. Routine examination of a variety of samples and organisms should be possible with this capability, addressing the need for a fast and comprehensive surveying tool with the capability to identify contaminants such as microplastics in complex backgrounds. The speed also provides an avenue for the emerging scientific area that seeks to comprehensively profile tissues via 3D reconstruction[33]. With this work, we should be able to reliably assess label-free histomorphology that utilizes tissue architecture and cell composition, providing insights into the structural and morphological details that are impossible with traditional IR microscopy and require extensive computation with other techniques. These advancements in speed, SNR, and segmentation are some essential steps in enabling the clinical translation of IR microscopy.

In this study, we have focused on attaining a level of performance that is suitable for most common biomedical tasks. If faster speed is needed, the base components of our design also offer an opportunity for further improvements. Complex tri-mirror scanner designs, for example, can allow for more homogenous illumination, greatly expanding the usable FOV of the system. Incorporation of faster scanning hardware—including resonant, polygon, piezoelectric, or MEMS-based devices—more efficient optical multiplexing[11], or higher-throughput FPGA-enhanced sampling schemes, can build upon the base design here to further increase both the speed and sensitivity.

Since conventional IR imaging systems are considerably slower compared to visible microscopy, they are not suitable for typical research or clinical samples. The potential of IR imaging for comprehensive histopathology has been proven in concept but a clinical evaluation to capture the diversity and true performance has been hampered by the need to measure a large cohort (~1000 patients) that is typical of definitive studies using AI in optical microscopy. Without a large-scale assessment of the accuracy of histopathologic models, tuning the performance of AI pipelines[52], and examining potential failure cases, a full assessment of the capability of IR histopathology was not possible. With the speed of IR-LSM, such assessments for many organs and diseases might now be possible. The improved throughput enables imaging larger data volume and more precise clinically relevant analysis. Furthermore, faster data acquisition can pave the way for making comprehensive training repositories publicly available. The faster speeds here can also help create curated datasets necessary for improving the integration of state-of-the-art AI engines with IR chemical imaging. Finally, the Food and Drug Administration (FDA) has created a pipeline to approve AI-based software as a medical device (SaMD) that necessitates incorporating high quality data and validation assurance. While traditional IR imaging instruments cannot acquire sufficient samples in a reasonable timeframe to establish clinically relevant AI models, our system can generate the large volumes of data necessary to develop AI engines that are qualified to be approved.

The design proposed here can also take advantage of emerging developments. QCLs are gradually becoming available beyond the mid-IR fingerprint region, for instance, the cell-silent window (2000–2500 $cm^{-1}$) enabling metabolic imaging with vibrational probes[10], to the C-H and O-H stretch regions (2800–3500 $cm^{-1}$). The stability of lasers is improving, and aided by balanced detection schemes[53], will eventually enable better sensitivity to molecular configurations[54], particularly by means of controlling polarization to measure circular dichroism[55]. Super-resolution techniques, such as deconvolution or structured illumination are also implementable in our design. Thus, extensions of this base confocal LSM design can be the foundation enabling future IR chemical imaging methods.

## Methods

### IR-LSM design

The system is configured as a laser scanning confocal microscope as illustrated in Fig. 1a. An external cavity (EC) quantum cascade laser (QCL) array (LaserTune, Block Engineering) containing 4 separate tuners and spanning a wavenumber range of ~777–1904 $cm^{-1}$ is tuned to a specific band of interest by rotating a grating (G) and intensity modulated with a duty cycle of 4% and pulse repetition frequency of 1 MHz. A beam combiner (BC), with the aid of a 532 nm diode laser for guidance, ensures collinearity and directs the emission through an illumination aperture ($A_I$) to the imaging arm via the primary beam splitter (BS) (KBr; Spectral Systems) while residual light is blocked by a beam dump (BD).

In the imaging arm, the beam is steered by an XY galvanometer (galvo) optical laser scanner ($\theta_{XY}$) (6215H, Cambridge Technology), with the fast axis controlled by a symmetric modified triangular waveform for bidirectional raster scanning (Supplementary Fig. S8a), thus avoiding fly-back time and achieving a scan duty cycle of ~93%. The scans in the forward and reverse directions are aligned by continuously tracking the real-time position output monitor from the fast axis servo board with external zero-crossing threshold circuitry, digitizing this line trigger, and then further adjusting the data stream by the system response time. The beam is transmitted through a custom optical train that is corrected for 3rd order aberrations and designed to

illuminate a diffraction limited spot on the sample at all points in the FOV.

In the detection arm, a pinhole (100 μm diameter) is placed conjugate to the illumination focal spot in the sample plane and sized to its first minima, post-magnification, at the design wavenumber. Note that as the diffraction-limited spot size scales by wavenumber, the performance of an average-sized pinhole will not be optimal over the entire tunable spectrum. The filtered beam is then focused using a 50 mm reflected focal length off-axis parabolic mirror (OAPM) onto a TE-cooled MCT detector (PVMI-4TE-10.6, VIGO Photonics). The pre-amplifier is adjusted to a bandwidth of 15 MHz; thus, the detector is sampled with a 250 ns delay following each laser pulse. All analog data acquisition, galvanometer drive signals, digital triggering, and state monitoring, are synchronized by a data acquisition card (PCIe-6361; National Instruments) in conjunction with in-house microscope control software (C#.NET).

The software continuously reads out the buffered pixels and constructs image frames that are then stored in a circular frame history buffer (Supplementary Fig. S8b). The final image that is displayed or stored by the virtual frame grabber consists of the most recent frame ($F_t$) acquired at time $t$, co-averaged with the n latest frames (through $F_{t-n}$) stored in the buffer, where n is user selectable or can be automatically adjusted depending on the SNR of the laser, which can vary greatly from band to band. Real-time monitoring of the microscopy stage, laser, and other critical equipment will flush the buffer in the event of state change thereby preventing inadvertent blurring of the images. The software then constructs multispectral images by sequentially grabbing frames synchronized to the laser tuning to a user-determined set of wavenumbers. It is also capable of acquiring point spectra at any point within the field of view (FOV) at a rate of up to ~10 Hz by sweeping the QCL source. In either case, a spectral background is first measured on a blank substrate for power referencing and non-uniformity correction. The system acquires 1 pixel per laser shot, resulting in a default pixel rate of 1 MHz, adjustable (up to 2 MHz on current models) depending on the pulse-to-pulse stability of the specific QCL unit selected. At standard settings, a $500 \times 500$ px tile corresponding to a FOV of $1 \times 1$ mm$^2$ (10×/0.4 NA) or $0.5 \times 0.5$ mm$^2$ (20×/0.8 NA) has a frame rate of ~4 Hz. For large area coverage, multiple frames are stitched together by blending with a ~10% overlap, which is adjusted at run-time depending on the total size of the mosaic. During the scan, the software may also automatically correct for sample tilt and/or focus, thus reducing error especially over longer experiments (Supplementary Fig. S9).

## Custom lens system design

All the core optical components of the IR-LSM system—including both microscopy objectives (OBJ) (10×/0.4 NA, 20×/0.8 NA), the tube lens (TL) (200 mm effective focal length), and a scan lens (SL) (4X)— were custom designed in-house (Supplementary Figs. S3–6) with manufacturing, evaluation, and final assembly outsourced (PIKE Technologies). All optical assemblies were designed to be apochromatic for a spectral range of ~833–1920 cm$^{-1}$, corrected at 3 frequencies within the span. This pair of IR objectives were designed to be infinity-corrected and telecentric, conveniently matching the standard specifications of common visible microscopy objectives. Design concepts, either based on libraries of prior art of visible microscopy components, or initiated with pseudo-random seeding, were optimized according to defined constraints and then evaluated (Code V; Synopsys). Only air gapped designs were permitted as cemented surfaces were deemed to be impractical to implement for IR compatible materials. Furthermore, additional care was taken to evaluate the effects and to mitigate internal reflections across the design spectral range, exasperated with the use of high index of refraction (IOR) elements.

The lack of suitable IR materials and their properties, as illustrated by an Abbe diagram (refractive index vs. dispersion, Supplementary Fig. S2), is severely restrictive. Generalizing, designs often include complementary sets of crown glasses (low IOR, low dispersion) and flint glasses (high IOR, high dispersion) to correct for aberrations. When nearly all IR materials have high IOR and low dispersion, the ability to perform corrections is limited and will inevitably generate designs requiring many elements. In low magnification systems, Ge-based designs are optimal due to having the highest IOR and lowest dispersion (among acceptable materials as indicated in Supplementary Table S2), thereby resulting in easy to manufacture singlet lenses with low curvature. In high magnification microscopy systems, the small dispersion is no longer negligible. Ge, lacking corrective power, is the least optimal material and was consequently not selected for either objective. Instead, the objectives were designed using the materials with the highest dispersions. Notably, BaF$_2$ was selected in most designs for its relatively low refractive index and high dispersion, critical for reducing the number of lenses required, despite its slightly hygroscopic property and fragility which made manufacturing difficult.

The number of lenses, surface complexity, broad-band (BB) anti-reflective (AR) coatings, and materials indicated in Fig. 1a, b were all carefully optimized to achieve performance goals while also controlling tolerance sensitivity, machinability, and costs. Multiple designs needed to be produced according to the unique capabilities and design guidelines of each potential manufacturing partner. Finally, the IR-LSM microscope was spatially calibrated using various negative chrome on glass targets (II-VI Max Levy) including USAF 1951, Siemens star, grid distortion, and Ronchi gratings. Spectral performance validation was performed on 5 μm thick films of SU-8 polymer spun coat onto IR reflective substrates.

## Sample preparation and imaging

Wild type zebrafish embryos were cultured in E3 medium at 28.5 °C fixed in 4% paraformaldehyde at 4 °C overnight at 28, 52, 76, 100, and 124 h post fertilization. After fixation, the embryos were washed with phosphate-buffered saline (PBS) and water, embedded in HistoGel (Epredia), and processed in paraffin. Adult zebrafish (18 months) were euthanized by submersion in ice cold water, fixed in 4% methanol-free paraformaldehyde, then paraffin embedded. These samples were sagittally sectioned, and 5 μm thick sections were collected on either standard glass or IR reflective low-emissivity glass microscopy slides (MirrIR; Kevley). Colon surgical samples were obtained from Carle Health (Urbana, IL, USA), embedded in paraffin blocks, and 100 consecutive sections 5 μm thick prepared on low-emissivity glass microscopy slides. These paraffin-embedded sections were deparaffinized in a ~24 h hexane bath. Alternatively, fresh frozen prostate surgical sections were acquired from Mayo Clinic (Rochester, MN, USA), cryo-sectioned onto low-emissivity glass, sealed and transported to the University of Illinois at Urbana-Champaign (Urbana, IL, USA) with dry ice, stored in a −80 °C freezer, and allowed to equilibrate to room temperature in a low-vacuum chamber and/or under pure nitrogen purge immediately before imaging. For all samples, select adjacent sections were also prepared on traditional microscopy slides, deparaffinized in xylene, stained with hematoxylin and eosin (H&E), then imaged using a brightfield microscopy whole-slide scanner (Nano-Zoomer 2.0-HT; Hamamatsu).

Multispectral imaging on the IR-LSM system was performed with 2 coadditions per tile while using the 10×/0.4 NA and/or 20×/0.8 NA objectives, configured to 2 μm and 1 μm pixel sizes respectively. High contrast spectral bands, including 1079, 1658 and 1765 cm$^{-1}$, were combined to create false-color images with each band representing a different color channel in RGB space (MATLAB; MathWorks). The images of serial tissue sections were color normalized, aligned, and

constructed into 3D volumes for visualization (ImageJ, National Institutes of Health). The system was compared to HD FT-IR technology, represented by a 680-IR spectrometer paired with a 620-IR imaging microscope (Agilent Technologies), equipped with a 15×/0.62 NA reflective Schwarzschild objective, and a liquid nitrogen cooled MCT $128 \times 128$ px focal plane array (FPA) detector. Spectroscopic images were acquired at 1.1 µm/px, 8 cm$^{-1}$ resolution, apodized using a Blackman-Harris function, windowed to a range of 800–3950 cm$^{-1}$, and averaged over 16 coadditions (Resolutions Pro; Agilent Technologies). These $141 \times 141$ µm$^2$ tiles were stitched together with in-house software (MATLAB; MathWorks) and visualized in ENVI + IDL 4.8 (ITT Visual Information Solutions).

## Machine learning

A deep neural network implementing U-Net architecture was trained for semantic segmentation of IR-LSM multispectral images of frozen prostate tissue into three major histological units: benign, cancerous, and non-epithelial tissue. To create the dataset, regions of interest (ROIs) from eight biopsies were imaged using the 10×/0.4 NA objective at 2 µm/px for each of the following spectral bands, indicative of the Amide I and II vibrational modes: 1454, 1544, 1572, 1596, 1620, 1658 and 1680 cm$^{-1}$. These bands were selected in part for suitable SNR, spatial resolution, and contrast, but also to demonstrate real-world applicability where applications may be constrained by a reduced spectral range such that all bands are accessible by potentially just a single tunable laser module, thereby reducing total system costs and improving the feasibility of clinical translation. During training, the IR-LSM images were labeled by first identifying the histologic classes in brightfield images of serially sectioned H&E-stained tissue through consultation with board-certified pathologists. The annotations were then manually copied onto the IR-LSM data, serving as ground truth labels. The network was trained on $128 \times 128$ px patches, created from 368 patches of $256 \times 256$ px extracted from the annotated regions and each down-sampled by factor of 2. For validation, we selected unique $4 \times 4$ mm$^2$ regions from three samples that were excluded from the training set.

The U-Net architecture contains convolution layer kernel sizes of $3 \times 3$ px, except for the final layer where a $1 \times 1$ px kernel was used to produce the desired class probability maps. Batch-normalization after each convolutional operation was performed to accelerate training. The softmax function was combined with cross-entropy loss to guide optimization. The network was trained for 10,000 iterations using the Adam optimizer with a learning rate of 10$^{-4}$. It was implemented in PyTorch 1.3, CUDA 10.1, and Python 3.7.1.

## Statistics and reproducibility

Representative experiments in this manuscript demonstrate the utility of IR-LSM. The reproducibility of the instrument to collect substantively similar data is optimized over years of operation and finally examined in Fig. 1 using synthetic ISO standardized targets and methods designed for evaluating the performance of imaging systems. These metrics are reported as noise and apply toward the demonstrative biological samples. The observed variances of the biological measurements are consistent within the noise limit.

## Ethics statement

This study protocol was reviewed and approved by the University of Illinois Institutional Animal Care and Use Committee (IACUC) under protocol #22160 and use of de-identified tissues under Institutional Review Board approval #06684.

## Reporting summary

Further information on research design is available in the Nature Portfolio Reporting Summary linked to this article.

## Data availability

The design files and data sets generated in this study necessary to interpret, verify and extend the research in the article are available from the corresponding author upon request due to their large file size. Requests for proprietary databases must be directed to the owner, for instance, the Code V material database from Synopsys (Mountain View, CA, USA). Design parameters published in the Supplementary Files are not intended for manufacturing. The authors recommend the published design be considered an initial guideline and adapted according to the requestor's specific needs, budget, and risk tolerance, as well as the preferences and capabilities of the selected manufacturing partner(s).

## Code availability

The acquisition software for the microscope, post-processing, and analysis code is available from the corresponding author upon request. The requestor is responsible for procuring all necessary licenses for 3rd party libraries used.

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

## Acknowledgements

This work was supported in part by the National Institutes of Health (NIH) via grants R01EB009745, P41EB031772, R01GM142172, and R21CA263147 to R.B.; the Mayo Clinic Advanced Diagnostics Laboratory innovation project (ADL0047 to R.B.); and the National Science Foundation (NSF) Division of Ocean Sciences Postdoctoral Fellowship (2205819 to M.P.C.). The authors thank Andy Bean (PIKE Technologies, Madison, WI, USA) for his design consultations and work toward the fabrication of the custom optical assemblies; Block Engineering (Southborough, MA, USA) for QCL-related support; and the following corporations for donating educational software licenses: Synopsys (Mountain View, CA, USA) for Code V optical simulation, SciChart (London, UK) for real-time charting, and Syncfusion (Morrisville, NC, USA) for GUI components.

## Author contributions

K.Y. and R.B. conceived of and designed the microscope system. K.Y. optimized the custom compound lens assemblies, built the microscope, wrote the control software, characterized its performance, and performed image post-processing. I.S. led engineering design and operational support. K.F. annotated data sets and trained the deep learning model. M.P.C., W.M., and A.B. performed all the Zebrafish experiments. Y.T.L. and M.M. programmed microscope subroutines. R.J.H. designed and performed detector subsystem tests. K.Y., M.P.C., and Y.P. collected data and evaluated the system. A.C.O. and M.M. provided engineering technical support. G.C. provided FFPE colon surgical samples and J.C.C. provided frozen prostate surgical samples along with pathological consultations. K.Y. and R.B. led the writing of the manuscript with input from all authors.

## Competing interests

The authors declare no competing interests.
