## [Peer Review File · Nature Communications]

Infrared Spectroscopic Laser Scanning Confocal Microscopy for Whole-Slide Chemical ImagingREVIEWER COMMENTS

Reviewer #1 (Remarks to the Author):

Infrared Spectroscopic Laser Scanning Confocal Microscopy for Whole-Slide Chemical Imaging

Kevin Yeh^{1,2}, Ishaan Sharma¹, Kianoush Falahkheirkhah^{1,3}, Matthew P. Confer¹, Andres C. Orr¹, Yen-Ting Liu^{1,4}, Yamuna Phal^{1,4}, Ruo-Jing Ho^{1,2}, Manu Mehta², Ankita Bhargava⁷, Wenyan Mei^{8,9,10,11}, Georgina Cheng^{11,12}, John C. Cheville¹³, and Rohit Bhargava^{*,1,2,3,4,5,6,11}

REVIEW 10 MARCH 2023

To Authors and Editor

An original and breakthrough contribution, designed and executed methodically with vision, and elegantly reported. This paper is of exceptional standard, the work is original and new.

The aim of the work is to break through the current impasse to report realisation of a brand new rapid biomedical analysis method which has the capacity to revolutionise chemical mapping of biological samples and beyond that give rapid diagnostic of disease when coupled with AI data analysis.

The Authors have designed an elegant microscope to exceed the imaging speed and quality of state-of-the-art QCL (quantum cascade laser) microscopes in DF (discrete frequency) conditions as well as surpass the efficiency of Ft-IR (Fourier transform InfraRed) microscopes under hyperspectral conditions. The design, its physical realisation, and exciting demonstrations are reported here in achieving the outstanding performance metrics.

The design centres on a confocal laser scanning microscope (LSM) corrected for third-order optical aberration across the large mid-IR bandwidth (1-12 microns cf. visible ~ 0.35 microns). The challenge was that there are no COTS (commercial of the shelf) optical assemblies for the mid-IR and therefore the system realisation rested on the ingenuity of the team to design and build optimised components in-house. This willingness to tackle a lack of COTS components in the mid-IR, which has frustrated rapid development of this field, is exemplary. The Authors are to be congratulated for their innovativeness, vision and persistence and ultimate success.

The paper is elegantly written. It makes a powerful case at the start for the need for a viable mid-IR chemical imaging to open up the field of of biomedical tissue diagnostics and beyond. The false colours

seen in the exquisite biological imaging in this paper are actually due to numerical spectral data that can be of great sensitivity and specificity for instance for disease recognition .

To Authors

ABSTRACT

Well written as far as goes.

Suggest:

1) adding an indication of time of image acquisition. So could a minimum time of acquisition of a certain image size to a certain resolution with a certain SNR be estimated to guide Reader from the start of your paper? Later you state that IR-LSM has higher imaging throughput (pixels/time) than FT-IR microscopes, can you put a number on this at best performance? I can see from the way the Abstract is written that you have preferred to leave this detail to the body of the paper and preferred to couch the time assessment in 'what ifs' and 'maybes'. I understand the reticence to be definitive because the influences are multifactorial and in any case you are conservative writers. Still this is such a good paper, can you stick your necks out and give the field at least some evidence in the Abstract of a guesstimate of time for CI of what all your inventiveness may be projected to achieve out in the field. Also time to, assuming instant AI data interpretation of, for instance, a cancer diagnosis in a pathology laboratory using your microscope.

2) Reader needs guidance in the Abstract about the nature of the image acquisition in this paper and time to collect implications. Is the spectral collection across the lateral dimension of the tissue sample in mapping mode (achieve by rasting the source beam and using a conventional MCT (mercury cadmium telluride) detector) or in imaging mode (using a focal plane array type MCT detector to collect directly the pixelated image based on the averaged mid-IR spectral absorption of tissue captured per pixel).If rasting, then what are the implications for time to acquire the CI?

3) Later still you observe:

"A

33 typically accepted standard for WSI by optical microscopy

34 is to scan a 15 mm x 15 mm area in minutes. Our designed

35 system requires 2.8 min/band with the 10X / 0.4 NA
36 objective for 2 μm/pixel quality. Typically, less than 10
37 bands are needed for many applications in histopathology,
38 thus bringing the time for scanning within range of time39
sensitive histology tasks such as intraoperative tissue
40 assessment, whose potential we demonstrate in the next
41 section focused on potential for clinical diagnostics. For
42 research purposes, the speed here allows for large scale
43 data recording that is necessary to assess extensive regions
44 of the spectrum or acquire high volumes of data for
45 sophisticated approaches such as deep learning. Notably,
46 the tissue imaging capability here is 10-fold faster than the
47 previous fastest laser scanning method^{19,20}, without
48 compromising SNR of recorded data.

49 The data acquisition speed of IR-LSM not only enables”

Could a punchy number from this paragraph be used to make a statement in the Abstract. For instance:
“CI for subsequent AI diagnostic of an unstained tissue 15x15 mm² achievable in 20 minutes.”.

To Authors and Editor: I can imagine that the authors are loathe to usurp their punchlines in the paper
by placing them upfront in the Abstract. But I feel they should be encouraged to ‘stick their neck out’ to
draw attention to these results in the Abstract itself.

Afterall, the Authors have achieved disruptive CI time-of-acquisition raising viability of deployment in a
clinical setting and use of their microscope as a bio-laboratory mainstream tool.

4) You write “Ground-up lens design”. Readers (those in materials science) could take this to mean that
you performed grinding and of the lens materials and then sintered the various selected ceramic
particles into lens shapes. I suggest this phrase is better expressed as: “Bottom-up lens design”.

To Authors and Editor:

GLOSSARY

Please ADD (NEW) GLOSSARY.

There is a lot of abbreviations across this paper, probably necessarily (Authors to check). I advise an alphabetical list at start (after Abstract) of ALL abbreviations in the paper to succinctly define these. This will help Reader appreciation of the work enormously. Please note I have tried to include all your abbreviations but up to Authors to make this list exhaustive:

A, AFS, Aper, Asper, Assy, BB, BC, BS, CFS, DFIR, EC-QCL, FAST, FF-PE, FOV Ft-IR, FM, H&E, IR-SLM, Ip, J, M, MTF, MPx, NA, OAPM, OBA, PV-MCT, RMS, SG, SL, SU8, TL, TPI, U-Net, WSI,

If you do not include this upfront Glossary, then each Figure caption definitely requires its own Glossary to ensure Reader understanding.

To Authors:

INTRODUCTION

1)

Elegant script setting out past history and what's new here.

2)

Corrections:

[9] to noise ratio (SNR) that is critical for biomedical

[9] to noise ratio (SNR) which is critical for biomedical

[10] Artifact-free data, in contrast, is obtainable by

[10] Artifact-free data, in contrast, are obtainable by

[16] design a microscope that exceeds the imaging speed and

[16] design a microscope which exceeds the imaging speed and

[38] emitting a narrow-band beam tunable across the molecular

RESULTS

Innovative engineering design and execution, whilst considering costs. Authors have created a series of fully custom compound optics, designed in tandem, in this new, mid-IR wavelength range. Needs must, the authors had no choice because there are no optics out there, to date, for mid-IR which, as Authors state, encompasses 10x bandwidth of conventional visible optics and 10x wavelength.

To Authors and Editor: I commend these Authors' approach which is almost a textbook type 'new lesson' for us in this mid-IR field and contributes much to new optics' engineering going forward. Well done indeed.

Authors please attend to:

1)

[13] calculated the modulation transfer function(MTF) using—

Please add one sentence here, defining what a MTF is and what it does, for non-expert?

2)

15 (vi). First, note that the resolution of the FT-IR imaging

16 system in transfection sampling is not symmetrical as part

17 of the back aperture of the objective is used for illumination

18 via a folding mirror.

Please add one sentence here, verifying whether the IR-LSM microscope is also in transfection, or is it in reflection? Transmission? or what?

3)

[30] effects is increases in using a coherent laser source. The

Change to:

[30] effects is increased in using a coherent laser source. The

4)

58 3e allows the visualization of any cross section from this

59 volumetric data, allowing an appreciation of the three60

I suggest that you make a proper statement at the start of this paragraph that CI using IR-LCM is intrinsically a 2-D process i.e. a surface imaging process due to the very high extinction coefficient of tissue vibrational absorption in the mid-IR spectral region. Then you can explain that 3-D CI is necessarily based upon sequential acquisition of 2-D CI.

5)

75 whether IR-LSM can enable more rapid pathology.

76 Samples can be prepared using a cryostat microtome to

77 provide chemical-structural assessment from excision to

78 image that is suitable for settings in which rapid assessment

79 is needed, for example, intraoperative diagnoses. We

80 demonstrate this capability by imaging and analyzing

81 frozen prostate tissues from surgical resections.

It is unclear why they must be frozen, which takes up valuable time. Presumably this is due to increasing the Young's modulus of the tissue for ease of microtome slicing (a diamond knife). Again it is unclear if the mid-IR CI is done of deep-frozen tissue or of freshly-thawed frozen tissue.

Authors state:

We used fresh frozen samples

114 for imaging, which better conserve chemical information³⁴,

115 but require imaging in <1 hr to prevent extensive

116 degradation and conserving tissue for further analysis³⁵

To Authors and Editor

7)

Another important aspect of all of this is that you did not dry the samples. Normally for Gold Standard histopathology, the tissue is formalin dried and can be ethanol exchanged so that water is replaced by ethanol to accept the H&E dye staining. So essentially, in these Authors' work, the tissue was 'wet' even if the water was present as ice (which I am still unclear about whether the tissue was deep frozen when

imaged). Note of course we are not talking about water or ice as a condensed phase of water molecules hydrogen bonded together. Rather, the water molecules are hydrogen bonded into the biomolecules of the tissue and quite probably no liquid water as such, nor ice, occurs at all. If the authors agree with this analysis of their process then we should agree that this is a new aspect, and as such it should be highlighted in your text at this point in the paper. Reason being, it is new.

To Editor + Authors

DISCUSSION

1)

Why on page 6 is there the title: "Online Methods"? when in fact this seems to be the start of detailed description of the built equipment. Could a better more informative title be found by the authors and should not it be capitalised as it is not a subsection of the Discussion. Suggest: EXPERIMENTAL DETAILS.

2)

Figures 1, 2, 3, 4, all need more vertical space assigned to them in order to tease apart the individual parts of the Figures. These individual parts are so crushed together, with no 'light of day' inbetween as to detract from the Reader's appreciation and understanding of these important figures.

To Authors

FIGURES 1-4

1) Figure 1

To make Figure 1 more visual and understandable for the Reader, I recommend

a) Complete definitions to be given in Fig1 caption of all abbreviations in the images of Fig1 and all abbreviations in the caption of Fig1.

b) Thus, please define in alphabetical order at the end of the caption: A, Aper, Asper, Assy, BB, BC, BS, EC-QCL, Ft-IR, FM, IR-SLM, Ip, J, M, MTF, MPx, OAPM, OBA, PV-MCT, SG, SL, SU8, TL,

c) Any that you would prefer to define in the text, state in the caption which these are and where they are to be found in main text.

d) Please state somewhere make/model/wavelength range of QCLs used – this is still missing from your experimental information.

e) Please explain in caption, or make it more clear on the diagrams, where objective lens set ups in part (b) of Fig1 fit into the system in part (a) of Fig 1

2) Figure 4

(a) Figure 4 caption states:

Figure 4. Demonstration of prostate cancer detection in fresh frozen tissue

Also Authors state in main text:

[113]capabilities of IR imaging. We used fresh frozen samples

Are tissue samples still frozen during image acquisition?- it sounds like it. So if not then instead of 'fresh frozen tissue' please state 'freshly-thawed frozen tissue'.

(b) Regarding Figure 4 also, what is the relation between the histology sections H&E stained and adjudicated on by pathologist and the sections for IR. It may have been made clear in the main text. But please also add explanation to Figure 4 caption about this. So presumably the IR sections are unstained and are the mirror image slice of the histology stained sections. Presumably the IR sections were not dried but were frozen (in order to microtomb) and stored in a deep freezer (for how long) and then freshly thawed for the IR microscopy.

3) Figure 2

(a) Regarding the explanation of Figure 2 in the main text, Authors state:

We illustrate these potentials of IR imaging by analyzing

73 zebrafish embryos, which are an ideal test vector because

74 of their genetic similarity with humans, relative low-cost

75 husbandry, small size, and abundant availability

(b) I recommend that the typical lateral dimensions of the zebra fish are stated in your text, are these cm or microns long? This information helps Reader understanding (I think they seem typically to be ~ 1 mm long and 200 microns wide, from the imaging and mag. bars?).

(c) Also I recommend authors make some statement here about the epidermal mid-IR transparency of the zebrafish– therefore enabling chemical imaging of the interior organs- if you as the authors agree that this is what is happening?

(d) Authors are to be congratulated on the innovative chemical imaging of PTFE (polytetrafluoroethylene) contamination ('microplastic') of the zebrafish. Neat demonstration of versatility and promise of this imaging approach.

To Authors and Editor

Please can Authors add a CONCLUSIONS section.

To Authors and Editor

SUPPLEMENTARY INFORMATION

Please add a brief narrative at the start of this section SUPPLEMENTARY INFORMATION telling your Reader what is to come in this section and why you have felt it necessary to include the data. Also please state briefly the significance of each of the Figures S1 to Figure S12 so that the Reader is introduced upfront at the very least what to expect as far as the sequence of the S-Figures. Please also inform the Reader upfront that, for convenience, tabulated information is given within the Figures, and that each Figures comprises many parts. All of this probably seems obvious to the authors. But it is useful for the Reader to be informed of this at the start so they know what to expect.

Improve all captions. They are too terse and do not explain the SFigures in enough detail for full Reader understanding.

To Authors

1)

Figure S1

(a) What is PT2018 etc.?

2)

Figure S2

(a) please annotate the abscissa as 'Abbe number' and not 'Abbe value' as is the convention.

3)

Figure S3

(a) Figure S3 is not a figure it is solely a table. I find this is like an undergraduate mistake – is there a reason for this?

(b) Please either state the mid-IR ellipsometer make and model you use to acquire these data OR give references to websites where you found these data so that your Readers can check authenticity of your evidence.

4)

Figure S4

(a) Where did you buy/ how did you make: "Cleartran (CLRTRN) is a form of chemical vapor deposition (CVD) ZnS modified by a post-deposition hot isostatic process."

(b) If you made it, what was the make and model of CVD equipment

(c) "is a form of" - what form of ZnS (crystallographic phase? Space group? Amorphous?). More detail required – does not meet Reader scrutiny.

(d) Is "STOP" on the table referring to (1) on the diagram? Please clarify. What do you mean by STOP?

(e) Are solid bodies of these lenses actually BaF₂ ? and then you have deposited ZnS coating on all surfaces of these solid body lenses? – for what purpose? – is it for anti-reflection? – if so then please state this.

(f) Caption does not attempt to explain the two tables. Please label them and explain what they are in caption. There are several missing definitions – for instance which of the two curved surfaces of each lens has the radius Y ? and what is meaning of 3,4 in second table?

5)

Figures S4-S7

Please see comments for Figure S4 and correct/amplify accordingly.

6)

Figure 8

(a) State dimensions are in mm?

7)

Figure S12

(a) S12(b) please explain significance of colours of plots – are these at certain mid-IR wavelengths?

To Authors and Editor

REFERENCES

1)

These are generally in a mess. There is no continuity of whether the journal name is in full or abbreviated and whether there are full author lists or AN Author et al. Reference list needs attention.

Reviewer #2 (Remarks to the Author):

In this report, Kevin Yeh et.al. presented a custom designed IR laser scanning confocal microscope for whole slide chemical imaging. The main innovation is the design of new optical elements (scan lens, tube lens, and objective lens) to improve the resolution, SNR, and field of view of IR imaging. Compared to traditional FTIR imaging and existing confocal IR imaging, the imaging speed is improved by more than 10-fold, enabling whole slide imaging within reasonable timeframe (tens of minutes) that is suitable for clinical applications. While it is a conceptually simple idea, the implementation is significant and urgently needed for the field of direct IR imaging. The authors demonstrated the application of the confocal IR microscope in imaging three different samples (all thin sections, either frozen or paraffin embedded): zebrafish, paraffin embedded colon tissue, and cryosectioned prostate surgical tissue. Manual and

machine-learning based Segmentation are used to highlight areas with distinct spectral features. Overall, the manuscript is well written. My main concerns are:

1. Descriptions for many important technical details are unclear. For example, Details of spatial and spectral noise characterization are missing. 100% line is mentioned but it is unclear how it is calculated from the data. Same for spatial noise and SNR calculations. There are many spikes in the spatial noise data. Are those actual noise or artefacts? Why is the spatial noise comparable for FTIR vs IR-LSM, but spectral noise for FTIR is much higher? How is the error bar determined?
2. While the images shown are impressive, there is no quantitative comparison with existing IR imaging techniques. Many sections only have hand-waving claims of superiority, or even claims of capability that is not demonstrated. They either need to be substantiated or removed. For example, page 3 line 78 says “where the additional chemical data can be used to assess a variety of toxicities”. Since none of these is shown, I would suggest removing it. Page 3 line 115 claims that “although widefield cameras have been used to visualize large tissue sections rapidly, the images are affected by the tissue structure itself due to speckle”. If possible, a comparison of the same sample should be included to illustrate this point. Considering that commercial solutions for IR imaging is available (Spero microscope) and offers high-quality imaging results, it would be helpful to have a direct comparison of the images. Quantitative metrics (resolution, field of view, speed, level of artifact) should be provided. FTIR is compared, but only for benchmarking, not sample imaging. Widefield DF IR imaging would be a more appropriate target for comparison because the main innovation is speed.
3. In the U-Net training for prostate tissue, it seems that the annotation (Figure 4a) does not contain all stroma tissue. Is there a reason for that? Is the fine annotation also done by a pathologist? What is the accuracy of the segmentation using U-Net from the validation set? The authors claim “It is evident that rapid IR imaging coupled with machine learning can provide accurate segmentation of benign and cancerous tissue from frozen sections, without the need for stains or human assessment”, but no data is provided to support the claim.

Additional comments:

1. Some labels in Figure 1 do not match what is described. For example, BB is in the figure but not described (should be BD?). There is another BB shown later. It might be helpful to have the description in the figure caption.
2. For the SNR calculation, the caption indicates 3 bands but the figure description says 100 bands.
3. How is the false color generated in Figure 2C-D. Is each color corresponding to one IR band? How many bands are used? Similarly, it is unclear how the green PTFE is generated. Is it a distinct band? Why does the Amide I band highlight skeletal structure instead of all tissue?
4. There are quite a few typos in the manuscripts that should be cleaned up.

**Authors' Response to Reviewers of
"Infrared Spectroscopic Laser Scanning Confocal Microscopy for Whole-Slide Chemical Imaging"**

We greatly appreciate the reviewers' time and energy in providing comprehensive reviews that have greatly improved this manuscript.

Reviewer 1 General Comments

To Authors and Editor

An original and breakthrough contribution, designed and executed methodically with vision, and elegantly reported.

This paper is of exceptional standard, the work is original and new.

The aim of the work is to break through the current impasse to report realisation of a brand new rapid biomedical analysis method which has the capacity to revolutionise chemical mapping of biological samples and beyond that give rapid diagnostic of disease when coupled with AI data analysis.

The Authors have designed an elegant microscope to exceed the imaging speed and quality of state-of-the-art QCL (quantum cascade laser) microscopes in DF (discrete frequency) conditions as well as surpass the efficiency of Ft-IR (Fourier transform InfraRed) microscopes under hyperspectral conditions. The design, its physical realisation, and exciting demonstrations are reported here in achieving the outstanding performance metrics.

The design centres on a confocal laser scanning microscope (LSM) corrected for third-order optical aberration across the large mid-IR bandwidth (1-12 microns cf. visible ~ 0.35 microns). The challenge was that there are no COTS (commercial of the shelf) optical assemblies for the mid-IR and therefore the system realisation rested on the ingenuity of the team to design and build optimised components in-house. This willingness to tackle a lack of COTS components in the mid-IR, which has frustrated rapid development of this field, is exemplary. The Authors are to be congratulated for their innovativeness, vision and persistence and ultimate success.

The paper is elegantly written. It makes a powerful case at the start for the need for a viable mid-IR chemical imaging to open up the field of biomedical tissue diagnostics and beyond. The false colours seen in the exquisite biological imaging in this paper are actually due to numerical spectral data that can be of great sensitivity and specificity for instance for disease recognition

We greatly appreciate the careful reading by the reviewer and all advice and comments. We are deeply honored by his/her enthusiasm for the many years of work reported here.

Reviewer 1 Comment 1.1

ABSTRACT

Well written as far as goes.

Suggest:

1) adding an indication of time of image acquisition. So could a minimum time of acquisition of a certain image size to a certain resolution with a certain SNR be estimated to guide Reader from the start of your paper? Later you state that IR-LSM has higher imaging throughput (pixels/time) than FT-IR microscopes, can you put a number on this at best performance? I can see from the way the Abstract is written that you have preferred to leave this detail to the body of the paper and preferred to couch the time assessment in 'what ifs' and 'maybes'. I understand the reticence to be definitive because the influences are multifactorial and in any case you are conservative writers. Still this is such a good paper, can you stick your necks out and give the field at least some evidence in the Abstract of a guestimate of time for CI of what all your inventiveness may be projected to achieve out in the field. Also time to, assuming instant AI data interpretation of, for instance, a cancer diagnosis in a pathology laboratory using your microscope.

We thank the reviewer, and he/she is indeed correct. We deliberately did not make claims in the abstract as too many studies are now replete with adjectives such as "video rate", "real time", "ultrafast" etc. that cause confusion in readers but do not represent objective advances as they can be attained by clever combinations of experimental conditions. We preferred to remain exceptionally careful with our benchmarks, such as elapsed time, since they are influenced by factors that are easily controlled during the design process by forfeiting unreported features or not reporting

unfavorable trade-offs. However, we recognize the point made by the reviewer and are now including what we believe to be an average expected value, but still tunable over a large range depending on user priorities. We added details into the abstract as recommended by the reviewer. The abstract was also edited slightly for clarity and is reproduced below. The specific performance standard requested by the reviewer is highlighted.

“Chemical imaging (CI), especially infrared (IR) spectroscopic imaging, enables label-free biomedical analyses while achieving expansive molecular sensitivity. However, its slow imaging speed and poor image quality impede widespread adoption. We present a mid-IR microscope design that simultaneously provides high throughput recording, low spectral noise, and high spatial resolution for imaging whole slides. The bottom-up design of its compound optical train enables dual-axis galvo laser scanning of a diffraction-limited focal point over large fields of view with its interchangeable, infinity-corrected, high numerical aperture (NA), refractive objectives. Quantum cascade laser (QCL) illumination allows for rapid chemical mapping of full spectral ranges or only the discrete spectral frequencies necessary for analyses. We demonstrate whole-slide, speckle-free imaging in ~3 min per discrete frequency with a $2 \times 2 \mu\text{m}^2$ pixel size using our 10X / 0.4 NA configuration, and high resolution $1 \times 1 \mu\text{m}^2$ pixel capability with its 20X / 0.8 NA counterpart, both offering spatial quality at their respective theoretical limits with minimal optical distortion or speckle while maintaining high signal to noise ratios (>100:1). The data quality enables superior applications of modern machine learning, leading to a suite of capabilities not previously feasible –3D reconstructions using serial sections, comprehensive assessments of whole model organisms, and histological assessments of disease in a time comparable to clinical workflows. Distinct from conventional stained approaches that focus on morphological investigations or molecular immunostaining techniques that target specific proteins or genetic alterations, this development makes all-digital, label-free imaging of minimally processed tissue practical.

Reviewer 1 Comment 1.2

2) Reader needs guidance in the Abstract about the nature of the image acquisition in this paper and time to collect implications. Is the spectral collection across the lateral dimension of the tissue sample in mapping mode (achieve by rasting the source beam and using a conventional MCT (mercury cadmium telluride) detector) or in imaging mode (using a focal plane array type MCT detector to collect directly the pixelated image based on the averaged mid-IR spectral absorption of tissue captured per pixel). If rasting, then what are the implications for time to acquire the CI?

We modified the abstract as indicated below to emphasize we are scanning a focused laser beam in the spatial domain and multispectral data is collected sequentially, band by band. All estimations in the manuscript indicate time/band, or in Fig.1, time per 100 bands such that we can better compare to FT-IR on a similar time scale. To acquire a CI for AI diagnostics, we assume ~ 5-10 bands would be required, depending on model. All FT-IR comparisons are to an MCT FPA-equipped commercial microscope.

“...We present a mid-IR spectroscopic microscope that simultaneously provides the high-throughput, low-noise, and fine-resolution required for modern research interests. Bottom-up lens design of a compound optical train designed around dual-axis galvo laser scanning of a diffraction-limited focal point permits spatial quality at the theoretical limits while achieving large fields of view. High-intensity quantum cascade laser (QCL) illumination allows for rapid chemical mapping using only the discrete spectral frequencies necessary for downstream analyses. The data quality enables superior applications of modern machine learning...”

Reviewer 1 Comment 1.3

3) Later still you observe:
“A typically accepted standard for WSI by optical microscopy is to scan a 15 mm x 15 mm area in minutes. Our designed system requires 2.8 min/band with the 10X / 0.4 NA objective for $2 \mu\text{m}$ /pixel quality. Typically, less than 10 bands are needed for many applications in histopathology, thus bringing the time for scanning within range of time sensitive histology tasks such as intraoperative tissue assessment, whose potential we demonstrate in the next

section focused on potential for clinical diagnostics. For research purposes, the speed here allows for large scale data recording that is necessary to assess extensive regions of the spectrum or acquire high volumes of data for sophisticated approaches such as deep learning. Notably, the tissue imaging capability here is 10-fold faster than the previous fastest laser scanning method^{19,20}, without compromising SNR of recorded data. The data acquisition speed of IR-LSM not only enables...”

Could a punchy number from this paragraph be used to make a statement in the Abstract. For instance: “CI for subsequent AI diagnostic of an unstained tissue 15x15 mm² achievable in 20 minutes.”.

To Authors and Editor: I can imagine that the authors are loathe to usurp their punchlines in the paper by placing them upfront in the Abstract. But I feel they should be encouraged to ‘stick their neck out’ to draw attention to these results in the Abstract itself.

Afterall, the Authors have achieved disruptive CI time-of-acquisition raising viability of deployment in a clinical setting and use of their microscope as a bio-laboratory mainstream tool.

We have changed the text to reflect this recommendation. Please refer to the response for Reviewer 1 Comment 1.1 above.

Reviewer 1 Comment 1.4

4) You write “Ground-up lens design”. Readers (those in materials science) could take this to mean that you performed grinding and of the lens materials and then sintered the various selected ceramic particles into lens shapes. I suggest this phrase is better expressed as: “Bottom-up lens design”.

We have changed the text to reflect this recommendation. Please refer to the response for Reviewer 1 Comment 1.2 above.

Reviewer 1 Comment 2

To Authors and Editor:

GLOSSARY

Please ADD (NEW) GLOSSARY.

There is a lot of abbreviations across this paper, probably necessarily (Authors to check). I advise an alphabetical list at start (after Abstract) of ALL abbreviations in the paper to succinctly define these. This will help Reader appreciation of the work enormously. Please note I have tried to include all your abbreviations but up to Authors to make this list exhaustive:

A, AFS, Aper, Asper, Assy, BB, BC, BS, CFS, DFIR, EC-QCL, FAST, FF-PE, FOV Ft-IR, FM, H&E, IR-SLM, lp, J, M, MTF, MPx, NA, OAPM, OBA, PV-MCT, RMS, SG, SL, SU8, TL, TPI, U-Net, WSI,

If you do not include this upfront Glossary, then each Figure caption definitely requires its own Glossary to ensure Reader understanding.

We have added a comprehensive “Glossary” section to help the reader better understand the technical components, defining each abbreviation, and in some cases, including additional text for disambiguation. However, we suggest it be located at the beginning of the supplementary information as it is very lengthy. Please refer to the manuscript file for the glossary text. We seek the editor’s discretion where this glossary section should be inserted. We included a footnote after the first abbreviation in the introduction section directing the reader to the supplementary. The caption for fig. 1, where abbreviations are most prevalent, now also reminds the reader to reference the glossary. The body text has been re-edited to define critical abbreviations and acronyms where appropriate. Some redundant abbreviations were corrected as well.

Reviewer 1 Comment 3.1

To Authors:

INTRODUCTION

1)

Elegant script setting out past history and what’s new here.

We appreciate the reviewer’s comment.

Reviewer 1 Comment 3.2

2)

Corrections:

[9] to noise ratio (SNR) that is critical for biomedical

[9] to noise ratio (SNR) which is critical for biomedical

[10] Artifact-free data, in contrast, is obtainable by

[10] Artifact-free data, in contrast, are obtainable by

[16] design a microscope that exceeds the imaging speed and

[16] design a microscope which exceeds the imaging speed and

[38] emitting a narrow-band beam tunable across the molecular

We thank the reviewer for the careful reading and bringing these issues to our attention:

We believe that “the limitations of microscope designs and/or spatial coherence preclude rapid acquisition of high signal to noise ratio (SNR) **that** is critical for biomedical analysis” is correct as it implies that there exist some biomedical analyses where SNR is not critical. If we wrote “which”, it would imply that high SNR is required for all biomedical analyses, and this wouldn’t be true as prior studies have been performed with legacy IR microscopes.

We corrected the sentence to “Artifact-free data, in contrast, **are** obtainable by conventional Fourier transform IR (FT-IR) imaging systems.”

We believe that “Overcoming these limitations, our goal here was to design a microscope **that** exceeds the imaging speed and quality of current state-of-the-art QCL microscopes” is correct as exceeding the speed and quality of current state-of-the-art microscopes is a restrictive clause describing our goal.

We corrected the sentence to “It uses an assembly of four external cavity QCLs, emitting a narrow-band beam tunable across **the** molecular fingerprint spectral range.”

Reviewer 1 Comment 4.1

RESULTS

Innovative engineering design and execution, whilst considering costs. Authors have created a series of fully custom compound optics, designed in tandem, in this new, mid-IR wavelength range. Needs must, the authors had no choice because there are no optics out there, to date, for mid-IR which, as Authors state, encompasses 10x bandwidth of conventional visible optics and 10x wavelength.

To Authors and Editor: I commend these Authors’ approach which is almost a textbook type ‘new lesson’ for us in this mid-IR field and contributes much to new optics’ engineering going forward. Well done indeed.

Authors please attend to:

1)

[13] calculated the modulation transfer function(MTF) using—

Please add one sentence here, defining what a MTF is and what it does, for non-expert?

We appreciate the reviewer’s comments. We recognize that non-experts may not realize how the modulation of features we referenced in the description of the Siemens star relates to the MTF. Therefore, we also edited the following sentence for clarification.

“The concentric rings of features patterned with known circumferential frequencies quantifies resolving power by determining the spatial frequency where the intensity modulation in the image falls below a predetermined threshold, commonly ~25% to approximate the Rayleigh criterion. **This ability of an optical system to measure contrast is described as the modulation transfer function (MTF), which we calculate using the slant-edge method (ISO 12233) as shown in fig. 1d(iv)-(vi).**”

Reviewer 1 Comment 4.2

2)

“First, note that the resolution of the FT-IR imaging system in transfection sampling is not symmetrical as part of the back aperture of the objective is used for illumination via a folding mirror.”

Please add one sentence here, verifying whether the IR-LSM microscope is also in transfection, or is it in reflection? Transmission? or what?

We thank the reviewer for bringing to our attention that the operation mode of IR-LSM was not clear. We mentioned the geometry as “transflected light” during the description of fig. 1a. We emphasize that both FT-IR and IR-LSM systems are set up in transfection geometry and have edited the text as follows:

“First, note that the resolution of the FT-IR imaging system in transfection sampling configuration is not symmetrical as part of the back aperture of the objective is used for illumination via a folding mirror. Hence, this asymmetric performance is quantified with separate MTF curves for the horizontal and vertical directions. In addition to poor vertical resolution, a loss of mid-frequency response can be observed in the MTF curves due to the central obscuration of the reflective objective. We do not see any such degradation in IR-LSM, which is also operated in transfection geometry, due to its use of refractive optics where light is illuminated and collected using the full aperture.”

Reviewer 1 Comment 4.3

3)

[30] effects is increases in using a coherent laser source. The
Change to:

[30] effects is increased in using a coherent laser source. The

We thank the reviewer for the careful reading and have edited this sentence accordingly.

“Despite the high number of refractive elements, the system exhibits negligible internal reflections (ghosting and lens flaring) even though the risks of such effects are ordinarily increased when using a coherent laser source.”

Reviewer 1 Comment 4.4

4)

“...3e allows the visualization of any cross section from this volumetric data, allowing an appreciation of the three...”

I suggest that you make a proper statement at the start of this paragraph that CI using IR-LCM is intrinsically a 2-D process i.e. a surface imaging process due to the very high extinction coefficient of tissue vibrational absorption in the mid-IR spectral region. Then you can explain that 3-D CI is necessarily based upon sequential acquisition of 2-D CI.

We thank the reviewer for bringing this potential point of confusion to our attention. We have adjusted this paragraph to read as follows.

“The speed of IR-LSM allows the capability to create volumetric 3D reconstructions based on IR data, providing a path to study both the complex structural and correlated chemical changes during disease processes in three dimensions. Note that IR-LSM is still intrinsically a 2D imaging technique, requiring the sample to be serially sectioned, due to the very high extinction coefficient of tissue vibrational absorption in the mid-IR spectral region.”

Reviewer 1 Comment 4.5

5)

“...whether IR-LSM can enable more rapid pathology. Samples can be prepared using a cryostat microtome to provide chemical-structural assessment from excision to image that is suitable for settings in which rapid

assessment is needed, for example, intraoperative diagnoses. We demonstrate this capability by imaging and analyzing frozen prostate tissues from surgical resections.”

It is unclear why they must be frozen, which takes up valuable time. Presumably this is due to increasing the Young’s modulus of the tissue for ease of microtomb slicing (a diamond knife). Again it is unclear if the mid-IR CI is done of deep-frozen tissue or of freshly-thawed frozen tissue.

The tissue samples must be frozen to cut with a cryotome, as is standard sample preparation for fresh tissue. The alternative is formalin-fixed, paraffin-embedded (FFPE) processing and sectioning using a microtome. “Fresh frozen” refers to the process of flash-freezing fresh tissue, preserving chemical information lost in FFPE, and without damage to the tissue due to formation of ice crystals. Used by Mayo Clinic since 1950s, the process only takes seconds but is typically not used since frozen tissue is harder to handle and requires more expertise. Moreover, all tissues are stored as FFPE; hence, most places only do FFPE processing, which takes over 24 hrs. Specifically, we have followed the protocol at Mayo clinic, which is:

“During an operation, tissue is transferred to the frozen tissue lab directly from the operating room. There, it is placed on a freezing microtome machine where the bottom of the sample is frozen within seconds. A razor-thin slice of tissue is extracted from the frozen section, prepared on a slide and placed under the microscope for review”, from <https://www.mayoclinic.org/departments-centers/surgery/overview/frozen-section-pathology-lab> on 4/11/2023.

We modified the first mention of fresh frozen in the results section to clarify, transferring text from later in the paragraph. “Fresh frozen” is a standard term in histopathology that we are hesitant to alter. Thus, we clarify the definition below.

“We demonstrate this capability by imaging and analyzing “fresh frozen” prostate tissues from surgical resections. These are fresh tissue samples that are flash frozen and sectioned on site, which better conserves chemical information³³, but requires imaging in <1 hr when outside deep-freeze to prevent extensive degradation. Thus, the speed of the IR-LSM is essential to examining fresh frozen tissue.”

With corresponding modifications in the sentence below.

Additionally, our use of unstained fresh frozen samples for imaging preserves the tissue structure and chemistry for further downstream analyses³⁵, including immunohistochemistry and/or sequencing.

The samples are not frozen during image acquisition. The tissue is cut extremely thin (typically 5 μm), so it dries very quickly. We don’t believe that “freshly-thawed” is the correct term as the sections are already dry when leaving the Mayo frozen tissue lab. They are sealed and packaged with dry ice during transport, then stored at UIUC at -80°C. When needed, they are transferred to a low vacuum chamber for temporary holding as the slides equilibrate in minutes to room temperature. Then, they are loaded into IR-LSM which is housed in an enclosure purged with N₂ gas. There is little opportunity for condensation to form. The methods section was modified accordingly to clarify this.

“...fresh frozen prostate surgical sections were acquired from Mayo Clinic (Rochester, MN, USA), cryosectioned onto low-emissivity glass, sealed and transported to the University of Illinois at Urbana-Champaign (Urbana, IL, USA) with dry ice, stored in a -80 °C freezer, and allowed to equilibrate to room temperature in a low-vacuum chamber and/or under pure nitrogen purge immediately before imaging.”

Reviewer 1 Comment 4.6

Authors state:

We used fresh frozen samples

“...for imaging, which better conserve chemical information³⁴, but require imaging in <1 hr to prevent extensive degradation and conserving tissue for further analysis³⁵...”

Please see above Reviewer 1 Comment 4.5.

Reviewer 1 Comment 4.7

To Authors and Editor

7)

Another important aspect of all of this is that you did not dry the samples. Normally for Gold Standard histopathology, the tissue is formalin dried and can be ethanol exchanged so that water is replaced by ethanol to accept the H&E dye staining. So essentially, in these Authors' work, the tissue was 'wet' even if the water was present as ice (which I am still unclear about whether the tissue was deep frozen when imaged). Note of course we are not talking about water or ice as a condensed phase of water molecules hydrogen bonded together. Rather, the water molecules are hydrogen bonded into the biomolecules of the tissue and quite probably no liquid water as such, nor ice, occurs at all. If the authors agree with this analysis of their process then we should agree that this is a new aspect, and as such it should be highlighted in your text at this point in the paper. Reason being, it is new.

Fresh frozen sections dry essentially instantly as they are sectioned thin (5 μm). This flash freeze drying technique is not new, but its usage in the clinic is not as widespread as FFPE as it needs immediate imaging and analysis, which few places can afford. Thus, the FF procedure is only routine at a few large medical centers with excellent resources, such as Mayo Clinic. For IR imaging, this is relatively new as well because FF processed tissue sections degrade when kept at room temperature >1 hr. Hence, FT-IR imaging of larger tissue samples is still reliant on FFPE processing. IR-LSM enables imaging in minutes and thus, opens this mode of tissue processing to analyses. Our goal in this section of the manuscript was to highlight IR-LSM and its viability with newer FF processing techniques, which are favored for real-time clinical decision making, rather than FF itself. While we do agree with the points that the reviewer brought up regarding FF, these would be tissue type dependent and would require a separate investigation.

Reviewer 1 Comment 5.1

To Editor + Authors

DISCUSSION

1)

Why on page 6 is there the title: "Online Methods"? when in fact this seems to be the start of detailed description of the built equipment. Could a better more informative title be found by the authors and should not it be capitalised as it is not a subsection of the Discussion. Suggest: EXPERIMENTAL DETAILS.

We changed this title to "Methods" according to the author guide. This section also contains several subheadings separating the different aspects of this work. Our subheadings are formatted to be inline with the paragraph.

Reviewer 1 Comment 5.2

2)

Figures 1, 2, 3, 4, all need more vertical space assigned to them in order to tease apart the individual parts of the Figures. These individual parts are so crushed together, with no 'light of day' inbetween as to detract from the Reader's appreciation and understanding of these important figures.

We appreciate the reviewer's advice and have reformatted each figure to allow for more white space between key parts. We reformatted some SI figures similarly. All changes were re-arrangements only, the underlying data in all figures remain the same. Corresponding changes have been made to the captions as well. **Please refer to the manuscript file for the updated figures, SI figures, and captions.** We thank the reviewer as these have made the figures significantly more legible and understandable.

Reviewer 1 Comment 6.1

To Authors

FIGURES 1-4

1) Figure 1

To make Figure 1 more visual and understandable for the Reader, I recommend

- a) Complete definitions to be given in Fig1 caption of all abbreviations in the images of Fig1 and all abbreviations in the caption of Fig1.
- b) Thus, please define in alphabetical order at the end of the caption: A, Aper, Asper, Assy, BB, BC, BS, EC-QCL, Ft-IR, FM, IR-SLM, Ip, J, M, MTF, MPx, OAPM, OBA, PV-MCT, SG, SL, SU8, TL,
- c) Any that you would prefer to define in the text, state in the caption which these are and where they are to be found in main text.
- d) Please state somewhere make/model/wavelength range of QCLs used – this is still missing from your experimental information.
- e) Please explain in caption, or make it more clear on the diagrams, where objective lens set ups in part (b) of Fig1 fit into the system in part (a) of Fig 1

We thank the reviewer for the advice to clarify fig. 1 for readers.

a-c) As there are a lot of abbreviations, to avoid an excessively lengthy caption, we chose to include them all in the glossary and referenced the glossary in the figure caption. Please refer to the manuscript file for the glossary text. The fig. 1 caption has been rewritten as copied below.

“Design overview and technical evaluations of label-free IR-LSM. (a) Schematic of the IR-LSM based on a galvanometer scanning confocal design with refractive optical assemblies custom-built for broadband performance and coupled to an EC-QCL array spanning the mid-IR spectral range. (b) Cross-sections of the interchangeable 10X and 20X objectives with design specifications presented in the supplementary information (supp. figs. S3-4) in conjunction with the tube and scan lens (supp. figs. S5-6). Technical comparisons between FT-IR and IR-LSM configurations including: (c) system noise and SNR calculations (supp. fig. S9) with evaluations per (i) single pixel 100% spectral lines, (ii) spatial noise per wavenumber, (iii-iv) spectral and spatial noise obtained over time (IR-LSM: 100 bands, FT-IR: 4 cm⁻¹ spectral resolution); (d) spatial resolution demonstrated with (i) subjective images of a Siemens star target, and with (iv-vi) objective MTF curves evaluated respectively at discrete wavenumbers using a slant-edge target; (e) total field of view; and (f) single pixel measurements of absorbance and derivative spectra of SU-8 photoresist deposition. A comprehensive glossary of terms is in the supplementary information.”

d) The identification of the QCL used is already included at the beginning of the “IR-LSM design” subsection within the Methods section.

“An external cavity (EC) quantum cascade laser (QCL) array (LaserTune, Block Engineering) containing 4 separate tuners and spanning a wavenumber range of ~ 777 to 1904 cm⁻¹...”

e) Fig 1 has been formatted to clarify where the detail of the objectives in (b) fits into the overall illustration in (a).

Reviewer 1 Comment 6.2

2) Figure 4

(a) Figure 4 caption states:

Figure 4. Demonstration of prostate cancer detection in fresh frozen tissue

Also Authors state in main text:

[113]capabilities of IR imaging. We used fresh frozen samples

Are tissue samples still frozen during image acquisition?- it sounds like it. So if not then instead of ‘fresh frozen tissue’ please state ‘freshly-thawed frozen tissue’.

(b) Regarding Figure 4 also, what is the relation between the histology sections H&E stained and adjudicated on by pathologist and the sections for IR. It may have been made clear in the main text. But please also add explanation to Figure 4 caption about this. So presumably the IR sections are unstained and are the mirror image slice of the histology stained sections. Presumably the IR sections were not dried but were frozen (in order to microtome) and stored in a deep freezer (for how long) and then freshly thawed for the IR microscopy.

While flash-freezing of fresh tissue is a newer process (relative to FFPE), it has been used in pathology practice since the 1950s. Hence, we refrain from introducing unique terminology and follow the standard pathology terminology. Fresh frozen refers to the method of preparation, not the state of the tissue during analysis which is always at room temperature (especially if stained). It is not unlike FFPE terminology, an industry standard description of only the method of preparation. We decided to clarify in the caption that these are fresh frozen tissue sections, and that we are not imaging the bulk tissue which must remain frozen.

“Demonstration of prostate cancer detection in a preparation of fresh frozen tissue sections. (a) Rough annotations guided by pathologists are (b) transferred from H&E images to IR images on consecutive sections. (c) A deep learning model is trained to predict epithelial (epi.) malignancy using multispectral IR images. (d) Validation is performed on unstained samples containing benign glands, malignant glands, and benign glands surrounded by malignant tumors, each imaged with IR-LSM and represented in each row. (e) Target ROIs are shown on the Amide I map, representing a full microscopy slide, with (f) indicative IR-LSM 10X magnified field of view simulating real-time screening. (g-h) Comparative images of H&E-stained adjacent sections of the same tissue sample are shown with corresponding ROIs marked.”

(b) The H&E images are of adjacent sections of the same fresh-frozen tissue sample that have undergone additional H&E processing. The caption was edited as shown above. We do not have detailed information on how long the samples were stored in deep freeze, either before or after cryosectioning. We estimate under a month in our freezer, but we did not include this in the manuscript as we did not believe it to be relevant information. To our understanding, tissues can be preserved cryogenically practically indefinitely. The procedure we envision is that samples would be imaged immediately after being processed at the frozen tissue lab, with IR-LSM also located at the clinic, and thus would not need to be transported or stored in deep freeze. We believe our current actions, limited by logistics, are reasonably equivalent such that the results presented, and discussion are still applicable.

Reviewer 1 Comment 6.3

3) Figure 2

(a) Regarding the explanation of Figure 2 in the main text, Authors state:

“We illustrate these potentials of IR imaging by analyzing zebrafish embryos, which are an ideal test vector because of their genetic similarity with humans, relative low-cost husbandry, small size, and abundant availability.”

(b) I recommend that the typical lateral dimensions of the zebra fish are stated in your text, are these cm or microns long? This information helps Reader understanding (I think they seem typically to be ~ 1 mm long and 200 microns wide, from the imaging and mag. bars?).

(c) Also I recommend authors make some statement here about the epidermal mid-IR transparency of the zebrafish—therefore enabling chemical imaging of the interior organs- if you as the authors agree that this is what is happening?

(d) Authors are to be congratulated on the innovative chemical imaging of PTFE (polytetrafluoroethylene) contamination (‘microplastic’) of the zebrafish. Neat demonstration of versatility and promise of this imaging approach.

(a-b) The embryonic zebrafish (fig. 2a-b) are each approximately 2 mm long. The adult zebrafish (fig. 3a) is ~ 25 mm long. The following sentence was edited to include this information.

“In contrast to the embryonic stages (~2 mm), adult zebrafish (~ 25 mm), present a more realistic model as they have fully formed organs.”

The zebrafish samples are all sectioned laterally. The methods explain the FFPE and sectioning preparation. All samples in this study are sectioned, it is not possible to IR image fully intact zebrafish samples. The text refers to “IR chemical maps for a whole section of wildtype zebrafish...”

We further edited the text to clarify this for the embryos.

“We illustrate these potentials of IR imaging by analyzing sections of zebrafish embryos...”

We also edited the fig. 2 caption to clarify this.

“Imaging sagittal sections of embryonic and adult zebrafish shown in false-color compositions of key IR absorption bands.”

(c) The zebrafish epidermis is not mid-IR transparent. We prefer to leave this discussion until later (see Reviewer 1 Comment 4.4) when discussing the 3d reconstruction of tissue. There, we state that this is still intrinsically a 2D imaging technique due to high absorption of mid-IR light.

(d) We thank the reviewer for the comments. Investigations of microplastic contamination is recently a widely recognized problem, and studies using zebrafish models have been well published using a variety of traditional methods, typically fluorescence microscopy. We show in this report that IR imaging can spectrally differentiate PTFE from biological material, and this may serve as a starting point for subsequent investigations.

Reviewer 1 Comment 7.1

To Authors and Editor

Please can Authors add a CONCLUSIONS section.

We referenced the author’s guide and other articles published in the most recent edition of the journal. The common practice appears to be to state conclusions within the discussion section. We do not find a dedicated conclusions section in any of the examples but happy to add an additional summary section if the editor would like us to.

Reviewer 1 Comment 8.1

To Authors and Editor

SUPPLEMENTARY INFORMATION

Please add a brief narrative at the start of this section SUPPLEMENTARY INFORMATION telling your Reader what is to come in this section and why you have felt it necessary to include the data. Also please state stating briefly the significance of each of the Figures S1 to Figure S12 so that the Reader is introduced upfront at the very least what to expect as far as the sequence of the S-Figures. Please also inform the Reader upfront that, for convenience, tabulated information is given within the Figures, and that each Figures comprises many parts. All of this probably seems obvious to the authors. But it is useful for the Reader to be informed of this at the start so they know what to expect.

Improve all captions. They are too terse and do not explain the SFigures in enough detail for full Reader understanding.

To Authors

1)

Figure S1

(a) What is PT2018 etc.?

We have added the following prelude to the supplementary information along with a list of supplementary figures and tables with brief description such that readers will be aware of what to expect. All captions throughout the supplementary information section have been rewritten and significantly expanded for better clarity. Please refer to the supplementary information file for the full text.

“This supplementary information contains a selection of reports listing engineering specifications and design performance targets for IR-LSM intended as an aid for expert readers to design similar mid-IR spectroscopic instruments in-house. As commercial off the shelf components are not available, and similar IR components (originally designed for surveillance or laser drilling and welding) will have subpar properties for microscopy while also being obscured in proprietary information, we have gathered and summarized key data in tabulated form. Additional information is available from the corresponding author upon reasonable request. We also

describe objective methods for system evaluation using widely available standardized targets, should readers wish to themselves compare the performance of IR-LSM with the field. We begin this supplementary information with a comprehensive glossary of terms and abbreviations.”

We have identified these systems instead as DFIR SSM and DFIR WFM with citation to publication. SSM and WFM are defined in the supplementary as stage scanning microscopy and widefield microscopy. These two systems represent the two direct predecessors of IR-LSM where we relied completely on commercial of the shelf components. The inclusion of these systems in this table, alongside FT-IR, demonstrates the impact of our shift to fully custom engineering.

Reviewer 1 Comment 8.2

2)
Figure S2
(a) please annotate the abscissa as ‘Abbe number’ and not ‘Abbe value’ as is the convention.

We have corrected this to match convention.

Reviewer 1 Comment 8.3

3)
Figure S3
(a) Figure S3 is not a figure it is solely a table. I find this is like an undergraduate mistake – is there a reason for this?
(b) Please either state the mid-IR ellipsometer make and model you use to acquire these data OR give references to websites where you found these data so that your Readers can check authenticity of your evidence.

(a) We apologize for this oversight and the error has been corrected.

(b) We did not measure these values ourselves. The purpose of this table is to supply refractive index information in a consistent format, focused on our target spectral range (5-13 μm), and calculated to approximately the maximum number of wavelength entries supported by Code V. This information is in the Code V material database and cross-checked with the materials’ suppliers. It is important that the reader confirms all material properties with the specific suppliers they intend to source material from as it may vary slightly with what is published and what can be found online. The allowable tolerance in these values depends on the optical design. We add the following into the caption of supp. table S2.

“These values can be derived from the Code V material database; however, verified material properties provided by the manufacturers should be used for accurate results.”

Reviewer 1 Comment 8.4

4)
Figure S4
(a) Where did you buy/ how did you make: “Cleartran (CLRTRN) is a form of chemical vapor deposition (CVD) ZnS modified by a post-deposition hot isostatic process.”
(b) If you made it, what was the make and model of CVD equipment
(c) “is a form of” - what form of ZnS (crystallographic phase? Space group? Amorphous?). More detail required – does not meet Reader scrutiny.
(d) Is “STOP” on the table referring to (1) on the diagram? Please clarify. What do you mean by STOP?
(e) Are solid bodies of these lenses actually BaF₂ ? and then you have deposited ZnS coating on all surfaces of these solid body lenses? – for what purpose? – is it for anti-reflection? – if so then please state this.
(f) Caption does not attempt to explain the two tables. Please label them and explain what they are in caption. There are several missing definitions – for instance which of the two curved surfaces of each lens has the radius Y ? and what is meaning of 3,4 in second table?

(a-c) Cleartran is a trademarked brand name, we did not make it. Similar material is referred to by different names, different grades, and can be sourced through many suppliers. We changed the manuscript throughout to refer to it as

multispectral ZnS (MS-ZnS). We sourced all material through PIKE Technologies as explained in the methods. The purpose of specifying the manufacturing processes is to disambiguate MS-ZnS (also known as Cleartran, clear-grade ZnS, or other essentially identical off-brand terminology) from other grades of ZnS which have very different optical properties. We decided to remove all mention of the MS-ZnS manufacturing process and rely on the glossary to provide disambiguation. We also edit the captions to state:

“Multi-spectral ZnS is also known as Cleartran, an optically clear grade of ZnS.”

(d) STOP refers to an aperture stop. We changed the notation of STOP to A_s , matching the notation in fig. 1 and this is also reflected in the glossary.

(e) The substrates used in this 3-element lens illustrated are MS-ZnS, BaF_2 , and MS-ZnS respectively. We did not specify the properties of the AR coating in the manuscript other than its wavelength range as we did not design it and this information is proprietary and manufacturer specific.

(f) These lens specifications tables are standard formatting, but we recognize the confusion and have modified the formatting slightly to clarify its meaning to more readers. The caption of fig. S3 was also rewritten to explain this convention.

“Optical design of the 10X / 0.4 NA microscope objective with each surface type, radius of curvature, surface thickness, and material as indicated. Tabular notation is as follows: surface 1 is represented by the aperture stop (A_s) followed by an air gap of 7.559 mm measured to the vertex of the next surface; surface 2 is the left surface of a MS-ZnS lens with a spherical radius of curvature of 14.562 mm and a substrate thickness of 4.744 mm; surface 3 is the right-side aspherical surface of this lens with a base radius of curvature of 31.076 mm, followed by an air gap of 2.448 mm. This aspheric surface 3 is of the form referenced in the glossary as *, with conic and aspheric coefficients as listed. Subsequent surfaces follow the same convention. The simulated MTF and CFS curves are evaluated at 1250 cm^{-1} and can be compared to experimental results in fig. 1d and supp. fig. s11a respectively.”

Reviewer 1 Comment 8.5

5)
Figures S4-S7
Please see comments for Figure S4 and correct/amplify accordingly.

All of these captions were changed to reference fig. S3 above where the convention was described in detail.

“Optical design of the 20X / 0.8 NA microscope objective following the tabular convention as described in supp. fig. S3. The simulated MTF and CFS curves are evaluated at 1250 cm^{-1} and can be compared to experimental results for the overall system in fig. 1d and supp. fig. s11a respectively.”

Reviewer 1 Comment 8.6

6)
Figure 8
(a) State dimensions are in mm?

Added the following into the caption of (now) fig. S7:

“All dimensions are in mm.”

Reviewer 1 Comment 8.7

7)
Figure S12

(a) S12(b) please explain significance of colours of plots – are these at certain mid-IR wavelengths?

Added the following into the caption of (now) fig. S11:

“Each color indexed trace indicates the z-traversal beginning at different offsets from the true focus at $z = 0 \mu\text{m}$.”

Reviewer 1 Comment 9.1

To Authors and Editor

REFERENCES

1)

These are generally in a mess. There is no continuity of whether the journal name is in full or abbreviated and whether there are full author lists or AN Author et al. Reference list needs attention.

We apologize for this oversight and the errors have been corrected per journal formatting. Please reference the manuscript file for the changes.

Reviewer 2 General Comments

In this report, Kevin Yeh et.al. presented a custom designed IR laser scanning confocal microscope for whole slide chemical imaging. The main innovation is the design of new optical elements (scan lens, tube lens, and objective lens) to improve the resolution, SNR, and field of view of IR imaging. Compared to traditional FTIR imaging and existing confocal IR imaging, the imaging speed is improved by more than 10-fold, enabling whole slide imaging within reasonable timeframe (tens of minutes) that is suitable for clinical applications. While it is a conceptually simple idea, the implementation is significant and urgently needed for the field of direct IR imaging. The authors demonstrated the application of the confocal IR microscope in imaging three different samples (all thin sections, either frozen or paraffin embedded): zebrafish, paraffin embedded colon tissue, and cryosectioned prostate surgical tissue. Manual and machine-learning based Segmentation are used to highlight areas with distinct spectral features. Overall, the manuscript is well written. My main concerns are:

We greatly appreciate the careful reading by the reviewer and all advice and comments. While this is conceptually a simple idea on its surface and has widely been recognized in literature as a potential solution that would address key concerns in DFIR imaging, the complexity was such that no research group or corporation offered a solution. We thank the reviewer in his/her recognition that this development is urgently needed for the field of direct IR imaging.

Reviewer 2 Comment #1

1. Descriptions for many important technical details are unclear. For example, Details of spatial and spectral noise characterization are missing. 100% line is mentioned but it is unclear how it is calculated from the data. Same for spatial noise and SNR calculations. There are many spikes in the spatial noise data. Are those actual noise or artefacts? Why is the spatial noise comparable for FTIR vs IR-LSM, but spectral noise for FTIR is much higher? How is the error bar determined?

We thank the reviewer for his/her advice. A significant portion of the captions in the supplementary information was rewritten to better explain the technical details. **Please refer to the supplementary information file for the full text.** In the main text, we reference this by stating:

“We first evaluate the spatial and spectral performance of this microscope in fig. 1c(i)-(ii) with calculations of these benchmarks described by supp. fig. S9.”

While the caption of supp. fig. S9 has been significantly expanded to explain these methods in detail. The caption of fig. 1 references this as well.

“Figure S9. Methods of evaluating system spectral and spatial noise, and corresponding SNR for figure 1c(i-iv) respectively. Collection of a hyperspectral dataset (A) with some empty background region (B) is required. The absorbance is calculated at each wavenumber as $A = -\log_{10}(I/I_0)$ by measuring the intensity of the light transmitted through the sample (I) and the original intensity of the laser (I_0). The resulting dataset is represented as $A[x, y, \nu]$ where $[x, y]$ are spatial coordinates and $[\nu]$ is the spectral coordinate. (a) The 100% line represents the spectral noise of the system upon detection of 100% of the available light, hence $I = I_0$ and $A = 0$, and is the point spectrum of a single pixel $[x, y]$ located within B, an empty region of the substrate. (b) The spatial noise is calculated from each acquired image per wavenumber. For each spectral band, the standard deviation of the set of pixels $\{x, y\}$ within B is the RMS noise of the system. (c) The spectral SNR is the signal-to-noise ratio calculated from the 100% line. The convention used in fig. 1c(iii-iv) is that the signal is 1 absorbance unit, common for IR spectroscopy, hence the spectral SNR is the inverse of the RMS noise calculated from a single pixel. Calculations are performed at different levels of coadditions (averaged replicate measurements), which are approximately proportional to time and to the square of SNR. Error bars represent one standard deviation from the mean of spectral SNR measurements each calculated from different sets of pixels $\{x, y\} \in B$. (d) Similarly, spatial SNR measurements are calculated from the spatial noise per wavenumber. Error bars are calculated as one standard deviation from the mean of spatial SNR measurements across bands, representing the spread of achievable SNR within the tuning range.”

The spikes in the spatial noise measurements for IR-LSM is explained in the text below. This presence of low power bands or noisy bands is unique to the laser system, whereas the noise for the FT-IR system is generally uniform, except for spikes associated with atmospheric water lines.

“The spatial noise in fig. 1c(ii) reflects the SNR of the measurement, showing a detector cutoff effect (lower wavenumber side) as well as higher noise for the LSM that is associated with low-power transitions of the multi-module laser system.”

The spatial noise is comparable for FTIR vs IR-LSM but only if the entire spectral range is being considered. For essentially all applications involving machine learning, only a small subset of wavelengths is required, hence the value proposition behind discrete frequency (DF) IR imaging. It is also essentially always possible to select nearby spectral frequencies that have low noise, thus these spikes in noise aren't impactful. We demonstrate this in all the images shown in this manuscript, where it is clearly apparent that the spatial noise of data collected with IR-LSM is much lower than that from FT-IR.

The spectral noise (100% line) of FT-IR with 16-coadditions is roughly equal to IR-LSM with 2-coadditions. These 100% lines are not normalized by time. This result is also shown in the Spectral SNR graph where their SNR is similar despite the FT-IR measurement taking substantially longer time. The errors bars (and mean) are determined by repeated measurements of the spectral and spatial noise. We added a detailed description of this, including other technical details regarding fig. 1, into the supp. fig. S9 caption copied above.

Reviewer 2 Comment #2

2. While the images shown are impressive, there is no quantitative comparison with existing IR imaging techniques. Many sections only have hand-waving claims of superiority, or even claims of capability that is not demonstrated. They either need to be substantiated or removed. For example, page 3 line 78 says “where the additional chemical data can be used to assess a variety of toxicities”. Since none of these is shown, I would suggest removing it. Page 3 line 115 claims that “although widefield cameras have been used to visualize large tissue sections rapidly, the images are affected by the tissue structure itself due to speckle”. If possible, a comparison of the same sample should be included to illustrate this point. Considering that commercial solutions for IR imaging is available (Spero microscope) and offers high-quality imaging results, it would be helpful to have a direct comparison of the images. Quantitative metrics (resolution, field of view, speed, level of artifact) should be provided. FTIR is compared, but only for benchmarking, not sample imaging. Widefield DF IR imaging would be a more appropriate target for comparison because the main innovation is speed.

Figure 1 is mostly dedicated toward quantitative comparisons with FT-IR, with evaluations of noise in fig. 1(c), spatial resolution in fig. 1(d), FOV in fig. 1(e), and spectral fidelity in fig. 1(f). FT-IR is still the gold standard for mid-IR imaging, and as this is mature technology, we believe that all manufacturers of FPA-based FT-IR microscopes will have reasonably similar performance. We include a comparison of quantitative specifications with FT-IR and previous generations of DFIR imaging systems we have designed in supp. table S1. Fig. 1d(i) is a comparison of FT-IR imaging with IR-LSM using a standardized sample (Siemens star target), which we believe to be substantially more objective than comparisons of tissue images. The drastic difference shown here in fig. 1 will apply when imaging tissue samples too, hence we do not include comparative FT-IR images in figs. 2-4.

We appreciate the reviewer bringing to our attention where we mistakenly made a claim when we intended to propose a study that may find the IR-LSM useful. We edited the sentence to:

“Embryo development and growth of the whole organism can be recorded (fig. 2a), revealing details at a level comparable to brightfield images of H&E stains (fig. 2b), but with additional mid-IR spectral information implicating chemical differences associated with various biological systems.”

Regarding the reviewer's comments about speckle, we changed the sentence to below as it is not just speckle prevalent in WF systems, but also artificial contrast and fringing due to scattering.

“Although widefield cameras have been used to visualize large tissue sections rapidly, the images are affected by the tissue structure itself, for example speckle arising from the coherent scattering of laser light¹⁴.”

We added a citation for a paper¹⁵ that used two separate Spero systems for imaging colon tissue sections, similarly, and they wrote extensively regarding the issues coherence posed in their study, stating that “coherence effects on sample edges and the sample itself must be addressed in future studies to obtain not only tissue classification but also comparable spectral quality to FTIR”. However, we believe the suggestion to compare IR-LSM with other DFIR systems (e.g. a stage-scanning system called “LDIR” by Agilent, or the “Spero” widefield system by Daylight) by publishing data that we collect from all systems would not be appropriate here. All FT-IR data, regardless of manufacturer, is reasonably similar, and all brightfield images are reasonably similar too, hence, we included data from these model instruments to serve as ground truth. Due to the short history and high variance of DFIR results currently in literature, there is no reference system available. Therefore, we focus this paper on IR-LSM.

We have added to supp. fig. S1(c-d) the performance of IR-LSM when imaging a USAF 1951 target, **partially copied below**. A similar sample has also been published with other QCL microscopes. This is a standard target for evaluating all imaging systems, similar to the Siemens star target in fig. 1. The specifications are defined by the U.S. Air Force MIL-STD-150A standard, and it is widely accessible to everyone, so we believe it is the most appropriate way for readers to compare different microscopes.

We also cite in the following statement a publication²³ authored by Daylight Solutions using their Spero microscope showing both an image USAF 1951 target and of colon tissue.

“Although the use of widefield measurements can provide larger fields of view, this design is neither limited by the size of the detector array nor by speckle or shot noise when an incident beam illuminates a large sample area”

To the best of our knowledge, we are not aware of more recent publications showing single FOV diagnostic images of other widefield microscopes, or publications that indicate how they have resolved this problem. Furthermore, in a more recent publication¹⁰, also co-authored by Daylight Solutions, they compare a DFIR Spero image with FT-IR, and state: “We note the DFIR images have some speckle patterns under the widefield configuration. This can be potentially mitigated by spatial noise dephasing or point scanning system” which is the key improvement of IR-LSM: a point scanning system that from the user’s perspective, functions like a WF system. We believe that the limitations of all WF systems are widely known, have been repeatedly acknowledged by the company in literature, and are not in contention.

We want to emphasize that our goal is not a comparison of or with commercial instruments. In this manuscript, we write about the general principles that describe the advantages and disadvantages of an IR laser scanning microscope (LSM) vs an IR stage scanning microscope (SSM) vs an IR widefield (WF) microscope. In this context, the main innovation here is not just speed, but a balance of speed and image quality that makes IR-LSM better suited for broad adoption when compared to SSM (lower speed, excellent image quality) or WF (lower image quality, excellent speed). Note that these tradeoffs are not brand specific, the same balance of image quality, speed, and costs apply when comparing a widefield microscope vs a LSM for fluorescence imaging. Hence, we do not believe a direct comparison with other commercial DFIR systems is productive here.

Reviewer 2 Comment #3

3. In the U-Net training for prostate tissue, it seems that the annotation (Figure 4a) does not contain all stroma tissue. Is there a reason for that? Is the fine annotation also done by a pathologist? What is the accuracy of the segmentation using U-Net from the validation set? The authors claim “It is evident that rapid IR imaging coupled with machine learning can provide accurate segmentation of benign and cancerous tissue from frozen sections, without the need for stains or human assessment”, but no data is provided to support the claim.

Since there are a large number of pixels in an IR image, typically only a subset of the image is annotated with confidence as guided by a board-certified pathologist. This is the standard workflow for IR imaging. Other regions of the image may be left unannotated or partially annotated. By doing this, we can create a dataset that is both practical and efficient for further analysis, while still providing valuable insights into the underlying structure of the image.

Our intention was not to provide a clinical study of prostate cancer using this instrument. A far larger data set would be required, and we believe this would be better served in a study that applies the progress of this manuscript to prostate or other tissues. Here, our focus was to introduce a new instrument that could potentially enable the integration of chemical imaging in the clinic. We edited the following section to include the accuracy of the segmentation from the validation set.

“We illustrate the performance of our method under three different conditions (fig. 4d-f), where raw IR images are evaluated by our model to ascertain the presence of benign epithelial cells in normal tissue, whereas malignant tumors are clearly identified in cancerous tissue, subjectively in agreement with the corresponding parallel H&E images (fig. 4g-h). The performance of this model is calculated to have AUC ROC values of 0.932 (benign), 0.952 (malignant), and 0.968 (non-epithelium) estimated using 10,000 pixels for each class.”

Reviewer 2 Additional Comment #1

Additional comments:

1. Some labels in Figure 1 do not match what is described. For example, BB is in the figure but not described (should be BD?). There is another BB shown later. It might be helpful to have the description in the figure caption.

We apologize for the mistake and have added a glossary in the supplementary information for all terms. The beam block (BB) in the figure conflicted with broadband (BB) in the manuscript, so now the figure refers to beam dump (BD) as suggested by the reviewer.

Reviewer 2 Additional Comment #2

2. For the SNR calculation, the caption indicates 3 bands but the figure description says 100 bands.

We apologize for the mistake and have corrected it. For DFIR, scan time scales linearly with the number of bands. We reported benchmarks for 100 bands to plot the SNR calculations for DFIR on the same scale as FT-IR. The IR-LSM images only take ~20s/MPx/band at 8 coadditions whereas the FT-IR takes ~35min/MPx/band at 1 coaddition. Nearly all tasks will not require 100 bands, the IR-LSM images in figs. 2-3 only use 3 bands, while most classification tasks take < 10.

Reviewer 2 Additional Comment #3

3. How is the false color generated in Figure 2C-D. Is each color corresponding to one IR band? How many bands are used? Similarly, it is unclear how the green PTFE is generated. Is it a distinct band? Why does the Amide I band highlight skeletal structure instead of all tissue?

Yes, each color channel corresponds to a single band. This is explained in the methods below but now edited to clarify more. Please note that this is for visualization only, different bands may be used, and not necessarily just 3. However, RGB are the simplest representation and hence we used these. The selection depends on the sample being imaged and the purpose of this color scheme is purely for visualization.

“Absorbance at high contrast spectral bands, here 1079, 1658 and 1765 cm^{-1} , can be combined to create a composite color image with each band representing a color channel in RGB space.”

The PTFE green color is a mask generated by the 1213 cm^{-1} band with a high threshold, then overlaid on top of the Amide I band. The IR spectrum of PTFE is very distinct from tissue, with a strong peak at 1213 cm^{-1} , thus making the segmentation simple. The purpose of this demonstration was to show IR-LSM can differentiate biological from non-biological material via spectral information, and that this may serve as a starting point for subsequent investigations.

The Amide I band highlights all tissue, not skeletal structure. To clarify, we edited the caption as follows.

“Detection of contamination of PTFE microplastics on the zebrafish tissue sample. The Amide I absorbance plotted per the grey-level scale highlights the gills of the zebrafish with PTFE particles, detected by absorbance at 1213 cm^{-1} , shown in green color.”

Reviewer 2 Additional Comment #4

4. There are quite a few typos in the manuscripts that should be cleaned up.

We apologize for the typos and have thoroughly edited the manuscript again.

REVIEWERS' COMMENTS

Reviewer #1 (Remarks to the Author):

Your resubmitted manuscript is excellent apart from one aspect which I shall try to justify.

According to the ISO/BS standard BS-ISO standard: the three INFRARED spectral regions are:

near-infrared (0.78-3 microns)

mid-infrared (3-50 microns)

far-infrared (50-1000 microns)

(British Standards Institution, BS ISO 20473:2007

Optics and photonics. Spectral bands. 2007, checked 2015, BSI. p. 10.)

Many people reading your article will simply assume when you use the term 'infrared' that you only mean near-infrared. Laser physicists do not really appreciate that the mid-infrared opens up new possibilities compared to the near-infrared. Of course, we know that extinction coefficients are up to 100,000 less in the near-infrared (encompasses the overtone and combination band absorption spectral region) than in the mid-infrared where the fundamental vibrational absorption bands lie.

So my advice is to ensure in the title, abstract and throughout your manuscript you specify always that you are meaning the mid-IR and not just 'IR'. In any case the mid-infrared has become a hot new topic of late. So:

TITLE: Mid-infrared spectroscopic laser scanning confocal microscopy for whole-slide chemical imaging.

ABSTRACT:

Chemical imaging (CI), especially mid-infrared (IR) spectroscopic imaging---

INTRODUCTION:

However, this exciting potential remains locked by a rate of mid-infrared (IR) chemical imaging data ---

Here we report a re-imagining of the mid-IR microscope to address these limitations by designing custom optical components

I will leave you to correct the rest of the manuscript if indeed you buy into my philosophy.

Reviewer #2 (Remarks to the Author):

The authors have adequately addressed all my concerns/comments. The paper represents a significant advance in IR imaging and has huge potential in IR-based label-free pathology.

**Authors' 2nd Response to Reviewers of
"Infrared Spectroscopic Laser Scanning Confocal Microscopy for Whole-Slide Chemical Imaging"**

We greatly appreciate the reviewers' time and energy in providing comprehensive reviews that have greatly improved this manuscript.

Reviewer 1

Your resubmitted manuscript is excellent apart from one aspect which I shall try to justify.

According to the ISO/BS standard BS-ISO standard: the three INFRARED spectral regions are:
near-infrared (0.78-3 microns)
mid-infrared (3-50 microns)
far-infrared (50-1000 microns)

(British Standards Institution, BS ISO 20473:2007
Optics and photonics. Spectral bands. 2007, checked 2015, BSI. p. 10.)

Many people reading your article will simply assume when you use the term 'infrared' that you only mean near-infrared. Laser physicists do not really appreciate that the mid-infrared opens up new possibilities compared to the near-infrared. Of course, we know that extinction coefficients are up to 100,000 less in the near-infrared (encompasses the overtone and combination band absorption spectral region) than in the mid-infrared where the fundamental vibrational absorption bands lie.

So my advice is to ensure in the title, abstract and throughout your manuscript you specify always that you are meaning the mid-IR and not just 'IR'. In any case the mid-infrared has become a hot new topic of late. So:

TITLE: Mid-infrared spectroscopic laser scanning confocal microscopy for whole-slide chemical imaging.

ABSTRACT:

Chemical imaging (CI), especially mid-infrared (IR) spectroscopic imaging---

INTRODUCTION:

However, this exciting potential remains locked by a rate of mid-infrared (IR) chemical imaging data ---

Here we report a re-imagining of the mid-IR microscope to address these limitations by designing custom optical components

I will leave you to correct the rest of the manuscript if indeed you buy into my philosophy.

We greatly appreciate the advice and support provided by the reviewer. While we agree with the reviewer's sentiment, we found that replacing all "IR" with "mid-IR" made the manuscript more cumbersome and in some situations would create some unusual terminology, e.g. "mid-IR optics". We also considered using "MIR" abbreviations.

However, common terminology in the market used to specify IR optical components and sensors is typically short-wave IR (SWIR 0.7-2.5 μm), mid-wave (MWIR 3-5 μm), and long-wave (LWIR 8-12 μm). Often, the IR atmospheric window (5-8 μm) is purposefully blocked in products related to IR imaging. We do want to be careful that readers attempting to build a similar microscope do not inadvertently use MWIR-compatible components (our optics are designed for 5.2-12 μm). If stock components are the only available option, and broadband (2-12 μm) specifications are not available, we recommend LWIR specifications, which is a tolerable design trade-off that we have used in previously published work.

Therefore, to avoid potential reader confusion, especially for constructors, we elected to begin with a discussion of mid-IR spectroscopy to direct the reader to the intended IR range, and specifically refer to mid-IR only in the context of explicitly specifying a wavelength range, e.g. "mid-IR fingerprint region". Elsewhere though, we believe that "IR" is sufficient shorthand, especially later in the manuscript, and generic enough to avoid confusion between mid-IR/MIR and MWIR.

Reviewer 2

The authors have adequately addressed all my concerns/comments. The paper represents a significant advance in IR imaging and has huge potential in IR-based label-free pathology.

We greatly appreciate the advice and support provided by the reviewer.